# Epidemiological data of an influenza A/H5N1 outbreak in elephant seals in Argentina indicates mammal-to-mammal transmission

Marcela M. Uhart [1,2] ✉, Ralph E. T. Vanstreels[1], Martha I. Nelson[3], Valeria Olivera [4], Julieta Campagna [5], Victoria Zavattieri[5], Philippe Lemey [6], Claudio Campagna [5], Valeria Falabella [5] & Agustina Rimondi [4,7] ✉

H5N1 high pathogenicity avian influenza virus has killed thousands of marine mammals in South America since 2022. Here we report epidemiological data and full genome characterization of clade 2.3.4.4b H5N1 HPAI viruses associated with a massive outbreak in southern elephant seals (*Mirounga leonina*) at Península Valdés, Argentina, in October 2023. We also report on H5N1 viruses in concurrently dead terns. Our genomic analysis shows that viruses from pinnipeds and terns in Argentina form a distinct clade with marine mammal viruses from Peru, Chile, Brazil and Uruguay. Additionally, these marine mammal clade viruses share an identical set of mammalian adaptation mutations which were also present in tern viruses. Our combined ecological and phylogenetic data support mammal-to-mammal transmission and occasional mammal-to-bird spillover and suggest multinational transmission of H5N1 viruses in mammals. We reflect that H5N1 viruses becoming more evolutionary flexible and adapting to mammals in new ways could have global consequences for wildlife, humans, and/or livestock.

The emergence of H5N1 high pathogenicity avian influenza (HPAI) viruses from clade 2.3.4.4b in 2020 triggered numerous outbreaks in wildlife worldwide[1]. In Europe and southern Africa, impacts to wildlife were particularly severe in seabird colonies, with losses in the tens of thousands[2–5]. Many outbreaks have likely gone underreported in other regions where influenza surveillance in animals is limited[6]. In 2021–2022, the H5N1 HPAI 2.3.4.4b viruses spread to North America, further impacting wildlife, especially waterbirds and birds of prey[7] and reassorting with endemic strains[8,9]. The virus then spread to South America in 2022 via multiple introductions[10,11], causing large-scale mortality of seabirds, with an estimated death toll surpassing 650,000 individuals[11–15].

Until recently, it was generally considered that H5N1 HPAI infections in mammals were largely limited to terrestrial carnivores that consumed or otherwise interacted with infected birds[16–18], and these viruses generally showed limited airborne transmissibility in mammalian models[19–21]. During the 2021–2022 panzootic, H5N1 HPAI caused episodic mortality of pinnipeds and cetaceans in Europe[22,23] and North America[24–26], but it was only upon reaching the Pacific coast of South America that the virus demonstrated an ability to cause large-scale mortality in marine mammals[11,27]. More than 30,000 South American sea lions (*Otaria byronia*) died as the H5N1 virus spread along the coast of Peru and Chile in 2022–2023, with porpoises, dolphins, and otters also being affected in smaller numbers[11,13–15,27–29].

[1]Karen C. Drayer Wildlife Health Center, School of Veterinary Medicine, University of California, Davis, USA. [2]Southern Right Whale Health Monitoring Program, Puerto Madryn, Argentina. [3]National Center for Biotechnology Information, National Library of Medicine, National Institutes of Health, Bethesda, USA. [4]Instituto de Virología e Innovaciones Tecnológicas, INTA-CONICET, Buenos Aires, Argentina. [5]Wildlife Conservation Society, Argentina Program, Buenos Aires, Argentina. [6]Department of Microbiology, Immunology and Transplantation, Rega Institute, Laboratory for Clinical and Epidemiological Virology, KU Leuven, Leuven, Belgium. [7]Robert Koch Institute-Alexander von Humboldt fellowship, Berlin, Germany. ✉e-mail: muhart@ucdavis.edu; rimondia@rki.de

Following the southward spread along the Pacific coast of South America, H5N1 HPAI viruses were detected in sea lions at the southern tip of Chile in June 2023[29]. By early August, the virus was detected for the first time on the Atlantic coast, in a sea lion rookery off southernmost Argentina. Then, over the following weeks, the virus spread rapidly northward along Argentina's Atlantic coast, killing hundreds of sea lions along Argentina's shores[30], eventually reaching Uruguay[31] and southern Brazil[32].

Shortly thereafter, in October 2023, we recorded mass mortality in southern elephant seals (*Mirounga leonina*) at Península Valdés in central Patagonia, Argentina, with an estimated death toll surpassing 17,000 individuals[33]. In this study, we present epidemiological data and full genome characterization of H5N1 clade 2.3.4.4b viruses associated with the outbreak in elephant seals and with concurrent tern mortality. We analyze data from the Península Valdés event and prior reports to investigate potential pathways of H5N1 virus transmission among marine mammals and birds in South America and document a rapidly spreading H5N1 marine mammal clade carrying mammalian adaptation mutations of potential public health concern.

## Results

### Elephant seal mortality at Punta Delgada breeding colony, Península Valdés

The earliest observation of elephant seal mortality in Península Valdés was on 25-Sep-2023 when navy personnel at the Punta Delgada Lighthouse noticed an unusually high number of dead pups on the beach. On 10-Oct-2023, we surveyed the breeding colony at Punta Delgada (Península Valdés, Argentina), and counted 218 living and 570 dead pups (including weaners) (Table 1 and Fig. 1A). This represented a more than 70-fold increase in pup mortality rate compared to the prior years for which comparable data was available (71% in 2023 vs. 1% in 2013, 2015 and 2022). By 13-Nov-2023, only 38 pups survived (95% mortality). At least 35 subadult/adult seal carcasses were recorded in the area, whereas in previous years, even a single dead adult seal was a rare sighting. No unusual mortality was seen in juveniles, which began gathering in the usual numbers in November (Table 1).

The mortality event led to significant changes in the elephant seal social structure (Table 1), with a progressive replacement of mature alpha males by subadults and a rapid decline in the number of breeding females. This manifested as a patchy distribution of seals with scattered females without pups as well as abandoned and sick pups. By

13-Nov-2023, all breeding structure was dissolved. There were no harems, only 9 males (all subadults not associated with females) and 9 females (8 with pup and 1 pupless) amidst carcasses of elephant seals (Supplementary Fig. 1A, B).

The absence of large alpha male elephant seals to chase away perceived intruders resulted in a larger number of South American sea lions commingling (several were found dead) among breeding elephant seals at Punta Delgada (Table 2). This prompted agonistic interactions with nursing elephant seal mothers (Supplementary Fig. 1C) and attempts at sexual interactions with pups (Supplementary Fig. 1D). Other interspecies interactions included the scavenging of elephant seal carcasses by kelp gulls (*Larus dominicanus*) (Supplementary Fig. 1E) and the presence of living and dead South American terns (*Sterna hirundinacea*) amidst elephant seal carcasses (Supplementary Fig. 1F). Some terns showed neurological signs of disorientation, decreased fear response and difficulty/inability to fly, and were not in social groups as would be expected. The tern death toll increased over time to almost 400 dead birds (Table 2).

As per the temporal distribution of events, mortality of elephant seal pups peaked between 25-Sep-2023 and 10-Oct-2023, whereas the majority of terns died about three weeks later, between 3-Nov-2023 and 13-Nov-2023. This temporal delay also occurred in Argentina as a whole, with large-scale mortalities of sea lions (mid-August to late September 2023) and elephant seals (late September to mid-October 2023) preceding the large-scale mortality of terns (early to mid-November 2023).

### Clinical signs and post-mortem findings in elephant seals

Elephant seal pups showing clinical signs consistent with HPAI were seen during all field surveys in October and November 2023. Symptomatic pups were lethargic, had difficulties rolling or galumphing, and presented with labored breathing, nasal discharge, repetitive head or flipper movements, and tremors (Fig. 1B–D and Supplementary Movie 1). Most symptomatic pups were motherless and alone or close to other abandoned or dead pups. During one field survey, several pups were seen at risk of drowning with the incoming tide (Supplementary Movie 1). Ill and dead pups ranged in age from newborn to about 3 weeks old (i.e., about to wean). Some carcasses of freshly deceased pups showed foam or mucous nasal discharge (Fig. 1D), and abundant white foam drained from the sectioned trachea of one individual (Fig. 1E). It is unclear whether this was due to infection or

**Table 1 | Number of living and dead southern elephant seals (*Mirounga leonina*) at Punta Delgada breeding colony (Península Valdés, Argentina) during the 2023 mortality event (three site visits) compared to baseline data from previous years (one site visit, a census conducted the first week of October)**

| Status | Age class, sex, and male alpha status | Baseline (first week of October census) | | | Mortality event [a] | | |
|---|---|---|---|---|---|---|---|
| | | 5-Oct-2013 | 4-Oct-2015 | 5-Oct-2022 | 10-Oct-2023 | 3-Nov-2023 | 13-Nov-2023 |
| Living | Pups (nursing) | 536 | 534 | 647 | 218 | 10 | 8 |
| | Pups (weaners) | 81 | 70 | 58 | 17 | 30 | 30 |
| | Alpha males (subadult 4 or adult) | 12 | 21 | 18 | 4 | 1 | 0 |
| | Alpha males (subadults 1 to 3) | 2 | 1 | 0 | 13 | 2 | 0 |
| | Subordinate males (subadults 1 to 3) | 5 | 37 | 47 | 28 | 25 | 9 |
| | Adult females | 589 | 707 | 746 | 370 | 12 | 9 |
| | Juveniles | 1 | 0 | 0 | 2 | 91 | 390 |
| Dead | Pups (nursing) | 7 | 2 | 5 | 570 | NE [b] | NE [b] |
| | Pups (weaners) [c] | 0 | 0 | 0 | 0 | 2 | 2 |
| | Subadults/Adults [c,d] | 0 | 0 | 0 | 13 | 30 | 35 |
| | Juveniles | 0 | 0 | 0 | 0 | 0 | 0 |

It should be noted that prior to 2023, the Península Valdés population of southern elephant seals had been increasing by 1.0 to 3.4% per year[94]

Notes: [a]The earliest observation of elephant seal mortality at Punta Delgada was on 25-Sep-2023, but no counts are available; [b]Not estimated because many carcasses had been buried by sand or removed by tides; [c]Degraded carcasses were also counted, hence counts should be interpreted as overlapping/cumulative; [d]Age subclasses and sexes combined, since carcass decomposition precluded the determination of age subclass and sex.

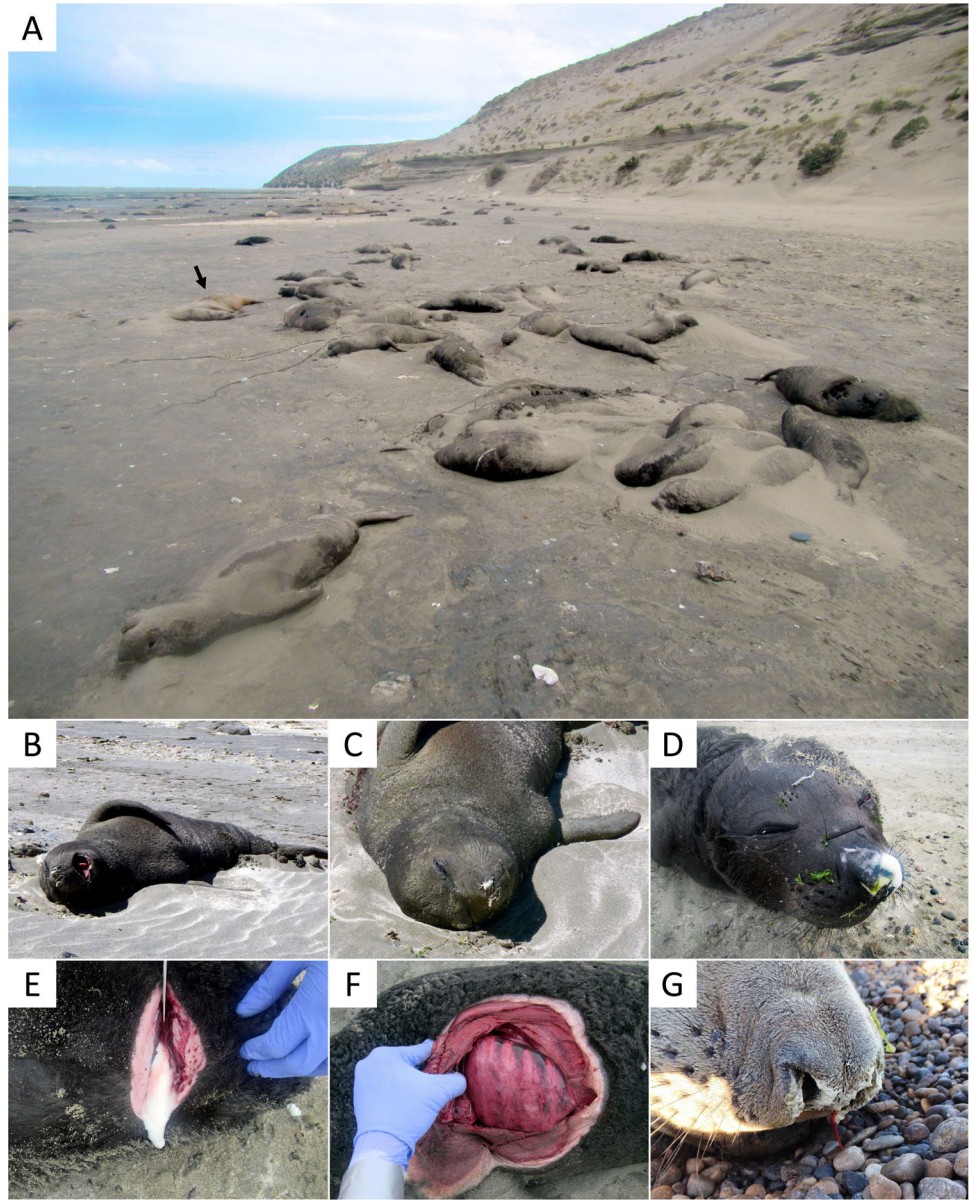

**Fig. 1 | Mass mortality, clinical signs, and post-mortem findings of elephant seals at Punta Delgada (Península Valdés, Argentina) during an outbreak of H5N1 HPAI. A** Hundreds of elephant seal pup carcasses accumulated along the high tide line of the beach at Punta Delgada; a sea lion carcass (arrow) and patchily distributed living elephant seals (far background behind the arrow) are also visible. **B** Pup presenting with open mouth breathing and tremors/twitching. **C** Pup presenting with labored breathing and foamy nasal discharge. **D, E** Abundant white foam on the snout and draining from the sectioned trachea of a dead pup. **F** Markedly heterogeneous and congested lung surface in a dead pup. **G** Bloody and mucous nasal discharge in a dead subadult male.

agonal drowning. The lungs of four pups showed a heterogeneous and congested surface (Fig. 1F), draining blood profusely when cut. We did not perform full necropsies due to biosecurity concerns; hence, we did not examine other organs. Following deaths in the breeding areas, several elephant seals hauled out at a second, aberrant location (Golfo Nuevo) in October-December 2023 (Supplementary Fig. 2 and Supplementary Table 1). Of these, one subadult male died within two days after showing clinical signs consistent with HPAI (tremors, labored breathing, yellowish and blood-stained nasal discharge, hyperthermia; Fig. 1G and Supplementary Movie 1).

### Detection of H5N1 HPAI clade 2.3.4.4b virus in wildlife at Península Valdés

We tested swab samples from four elephant seal pups, five South American terns, and two royal terns (pooled according to species and sample type), and an additional pool containing all samples from a sixth South American tern from Punta Delgada. Another pool containing all samples from a dead subadult male elephant seal from Golfo Nuevo was also analyzed. All pools were positive for the influenza A virus matrix gene (results detailed in Supplementary Table 2). Elephant seal samples were tested for H5 clade 2.3.4.4b and were positive, and sequencing was performed on individual samples where possible. In total, we performed whole genome sequencing of H5N1 HPAI viruses from three elephant seal pups (CH-PD027, CH-PD032, CH-PD035), one subadult male elephant seal (CH-PM053), two South American terns (CH-PD030 and CH-PD037), and one royal tern (CH-PD036). For one of the elephant seal pups (CH-PD032), we performed whole genome sequencing of viruses from five different samples (oronasal, tracheal, rectal, brain, lung), and nucleotide sequences from the eleven strains obtained were deposited in GenBank (Table 3).

**Table 2 | Estimated number of dead individuals of other pinniped species and seabirds at Punta Delgada (Península Valdés, Argentina) in 2023, during the elephant seal mortality event**

| Species | 10-Oct-2023 | 3-Nov-2023 | 13-Nov-2023 |
|---|---|---|---|
| South American sea lion (*Otaria byronia*) [a] | 20 | 4 | 8 |
| South American fur seal (*Arctocephalus australis*) [a] | 0 | 1 | 0 |
| South American tern (*Sterna hirundinacea*) [b] | c. 100 [c] | 178 [d] | 396 |
| Royal tern (*Thalasseus maximus*) [b] | 3 | 7 | 1 |
| Cayenne tern (*Thalasseus acuflavidus eurygnathus*) [b] | 1 | 2 | 2 |
| Kelp gull (*Larus dominicanus*) [b] | 3 | 10 | 15 |
| Imperial cormorant (*Leucocarbo atriceps*) [b] | 0 | 2 | 5 |
| Great grebe (*Podiceps major*) [b] | 0 | 1 | 3 |
| Peregrine falcon (*Falco peregrinus*) [b] | 1 | 1 | 1 |

Notes: [a] Counts of pinnipeds after the first visit affected by older carcasses being buried by sand or removed by tides. [b] Degraded carcasses were also counted, hence counts should be interpreted as overlapping/cumulative; [c] One live symptomatic individual seen; [d] Four live symptomatic individuals seen.

**Table 3 | H5N1 HPAI viruses from wildlife in Península Valdés, Argentina, 2023**

| Strain name | Host | Sample | Age | Collection date | Location | GenBank accession no. |
|---|---|---|---|---|---|---|
| A/southern elephant seal/Argentina/CH-PD027/2023 | *Mirounga leonina* | Rectal | Pup | 10-Oct-2023 | Punta Delgada | PQ002111–PQ002118 |
| A/southern elephant seal/Argentina/CH-PD032_oronasal/2023 | *Mirounga leonina* | Oronasal | Pup | 10-Oct-2023 | Punta Delgada | PQ002119–PQ002126 |
| A/southern elephant seal/Argentina/CH-PD032_tracheal/2023 | *Mirounga leonina* | Tracheal | Pup | 10-Oct-2023 | Punta Delgada | PQ002127–PQ002134 |
| A/southern elephant seal/Argentina/CH-PD032_lung/2023 | *Mirounga leonina* | Lung | Pup | 10-Oct-2023 | Punta Delgada | PQ002135–PQ002142 |
| A/southern elephant seal/Argentina/CH-PD032_brain/2023 | *Mirounga leonina* | Brain | Pup | 10-Oct-2023 | Punta Delgada | PQ002143–PQ002150 |
| A/southern elephant seal/Argentina/CH-PD032_rectal/2023 | *Mirounga leonina* | Rectal | Pup | 10-Oct-2023 | Punta Delgada | PQ002151–PQ002158 |
| A/southern elephant seal/Argentina/CH-PD035/2023 | *Mirounga leonina* | Brain | Pup | 10-Oct-2023 | Punta Delgada | PP488310–PP488317 |
| A/southern elephant seal/Argentina/CH-PM053/2023 | *Mirounga leonina* | Rectal | Sub-adult | 01-Nov-2023 | Puerto Madryn | PP488318–PP488325 |
| A/South American tern/Argentina/CH-PD030/2023 | *Sterna hirundinacea* | Brain | Adult | 10-Oct-2023 | Punta Delgada | PP488326–PP488333 |
| A/South American tern/Argentina/CH-PD037/2023 | *Sterna hirundinacea* | Pool | Juvenile | 10-Oct-2023 | Punta Delgada | PP488342–PP488349 |
| A/royal tern/Argentina/CH-PD036/2023 | *Thalasseus maximus* | Brain | Adult | 10-Oct-2023 | Punta Delgada | PP488334–PP488341 |

## Evolution of H5N1 HPAI viruses in Argentina

We first inferred a maximum likelihood tree for the HA segment to compare our H5N1 HPAI viruses in Península Valdés with other strains from South America, North America, and Eurasia during 2021–2023. This analysis confirmed that the H5N1 HPAI viruses detected in South America (and South Georgia) from November 2022 to November 2023 stem from a single introduction of clade 2.3.4.4b from North American wild birds (Supplementary Fig. 3). The H5N1 HPAI viruses in Argentina have the B3.2 genotype, including the eleven viruses sequenced for this study, six viruses sequenced from our previous report[30], and 46 viruses from poultry and one wild bird (Andean goose) available in GISAID (see Supplementary Fig. 3 and maximum likelihood trees provided in Zenodo[34]). The B3.2 viruses introduced from North America into South America have a reassortant genotype with four segments from the Eurasian H5 lineage (PA, HA, NA, and MP) and four segments from low pathogenicity avian influenza viruses from the North American lineage (PB2, PB1, NP, and NS)[9]. However, Argentina's H5N1 HPAI viruses are not monophyletic (i.e., clustering together as a single Argentina clade, separate from viruses from other South American countries). Instead, viruses collected from Argentina's inland poultry outbreaks are positioned in a different section of the tree from Argentina's coastal outbreaks in marine mammals and terns (Fig. 2A and Supplementary Fig. 4). Argentina's poultry viruses are positioned in the lower section of the tree along with poultry viruses from other South American countries (e.g., Uruguay and Chile), as well as some wild bird viruses from Uruguay, Brazil, Chile, Argentina's single H5N1

HPAI virus from an inland wild bird (A/goose/Argentina/389-1/2023; collected 11-Feb-2023), and wild bird and mammal viruses from South Georgia. Within this clade, Argentina's poultry viruses are intermixed with viruses from other locations and wild bird hosts, suggesting frequent virus movement across national borders and spillover between wild birds and poultry.

Conversely, the vast majority of wild bird viruses from Peru and Chile are positioned in the upper section of the tree in Fig. 2A. This wild bird clade is closely related (and basal) to a clade of marine mammal viruses collected from five countries (Peru, Chile, Argentina, Uruguay, and Brazil). A quantitative estimate of virus gene flow ("Markov jump" counts, Fig. 2B and Supplementary Fig. 5) indicates that H5N1 HPAI viruses transmitted approximately 4x from wild birds to marine mammals on the Pacific (western) coast of South America. Three wild bird-to-mammal transmissions in Peru appear to be dead-end spillover events with no secondary cases (A/common dolphin/Peru/PIU-SER002/2022, A/South American sea lion/Peru/AQP-SER00K/2023, and A/South American sea lion/Peru/LIM-SER036/2023) (Fig. 2A and Supplementary Fig. 6). Multiple independent wild mammal infections also occurred in South Georgia, although it is unclear whether onward mammal-to-mammal transmission was involved or not. In contrast, a single wild bird-to-marine mammal transmission that occurred on South America's west coast (likely Peru or Chile) in late 2022/early 2023 gave rise to a multinational clade of 40 viruses, including 33 from marine mammals in Peru (*n* = 2), Chile (*n* = 8), Argentina (*n* = 15), Uruguay (*n* = 6) and

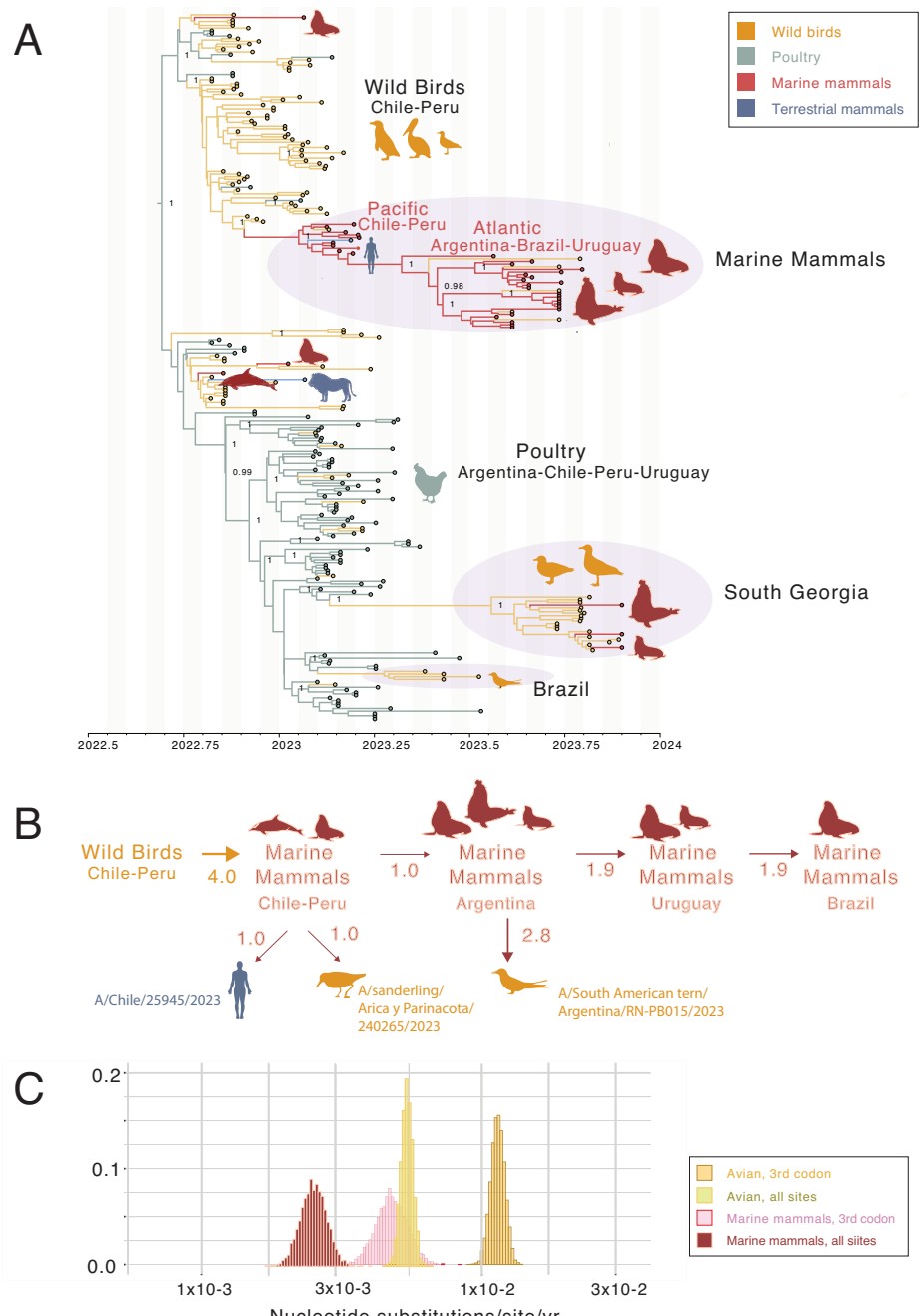

**Fig. 2 | Phylodynamics of H5N1 HPAI (2.3.4.4b) viruses in South American marine mammals and birds. A** Time-scale MCC tree inferred for the concatenated genome sequences (~13 kb) of 236 H5N1 influenza A viruses (clade 2.3.3.4b) collected in five South American countries (Argentina, Brazil, Chile, Peru, Uruguay) and in South Georgia and the Falkland/Malvinas Islands. Branches are shaded by inferred host species (4 categories). Posterior probabilities are provided for key nodes. The same tree with tip labels and posterior probabilities for all nodes is available in Supplementary Fig. 4. **B** Direction of virus gene flow between locations and hosts, inferred from "Markov jump" counts across the posterior distribution of trees inferred using a Bayesian approach (values under 0.5 excluded). Different host groups are indicated with different colors: wild birds (orange), marine mammals (red) and terrestrial mammal (blue). The same graphic with 95% HPD (highest posterior density) labels is available in Supplementary Fig. 5. Tree with all location states labeled is available in Supplementary Fig. 6. **C** Posterior distributions of evolutionary rates (substitutions per site per year) inferred for the complete virus genome (all positions) and for only the third nucleotide position for H5N1 (2.3.4.4b) in South America, partitioned into two host categories: marine mammal clade (excluding any human and avian viruses) and avian (wild bird/poultry) clade (excluding any mammal viruses). Source data for the histogram graph are provided as a Source Data file.

Brazil (*n* = 2). The marine mammal clade can be subdivided into two geographical groups of viruses: the earlier viruses collected in March – April 2023 on the Pacific (western) coasts of Peru and Chile, and the later viruses collected in August – November 2023 on the Atlantic (eastern) coasts of Argentina, Uruguay, and Brazil (Fig. 2A).

**Spillback from marine mammals to coastal birds and one human**
The multinational marine mammal clade also includes one human case from Chile (A/Chile/25945/2023), one wild bird virus from Chile (A/sanderling/Arica y Parinacota/240265/2023), one wild bird virus from the Falkland/Malvinas Islands (A/Southern fulmar/Falkland Islands/133789/2023) and four viruses obtained from terns in Argentina (one

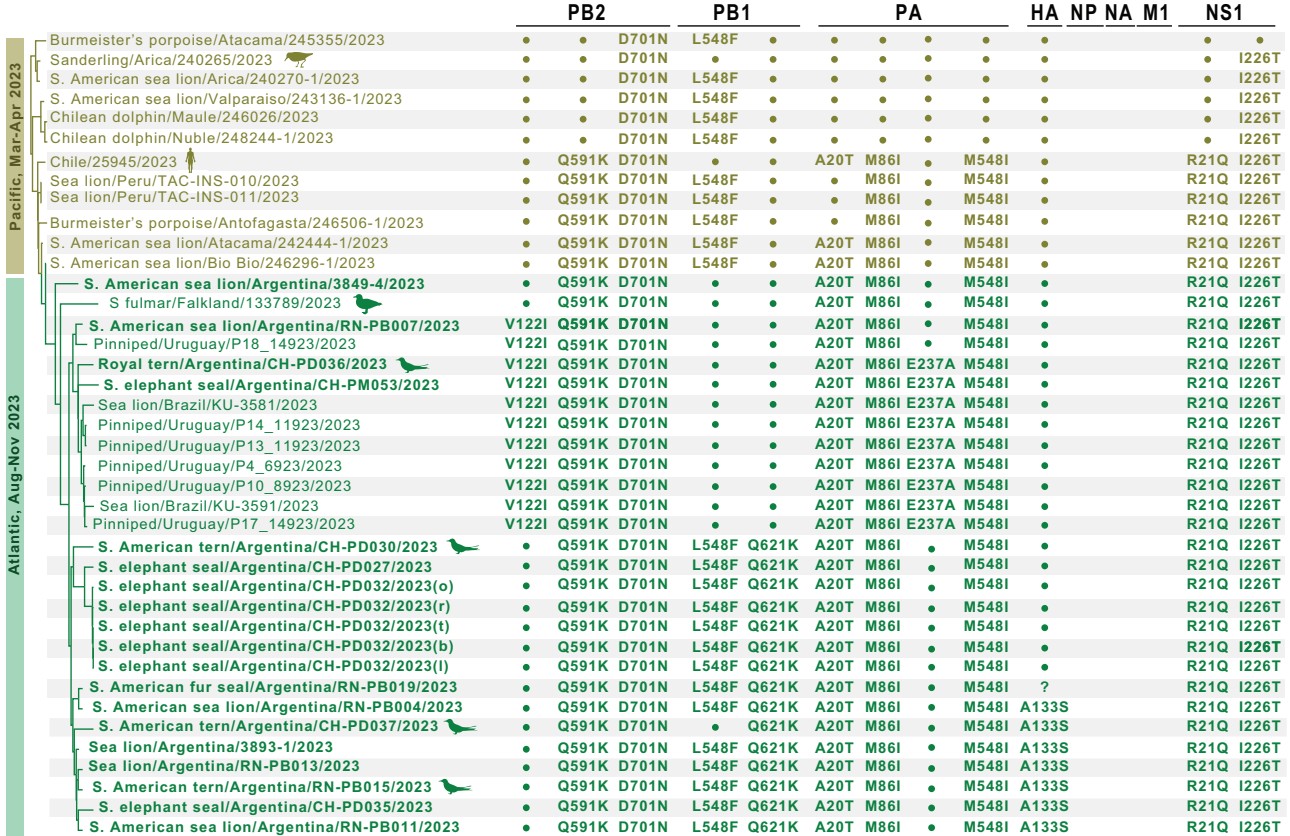

**Fig. 3 | Mutations defining the marine mammal clade of H5N1 HPAI (2.3.4.4b) viruses.** Amino acid changes are listed for new mutations that arose in the marine mammal clade of the H5N1 HPAI (2.3.4.4b) viruses that are not observed in any other avian viruses included in this study from South America, mapped against the subsection of the MCC tree with the marine mammal clade (see Fig. 2A). Virus names and associated mutations are colored by country. A question mark indicates that no sequence data is available at that position for that virus. HA mutations refer to H5 numbering.

South American tern from Punta Bermeja in August 2023, one royal tern and two South American terns from Punta Delgada in October 2023). The four tern viruses from Argentina are closely related to the marine mammal viruses from Argentina but appear to be independent mammal-to-bird spillbacks, with no clear evidence of tern-to-tern transmission. The human and sanderling viruses positioned in the marine mammal clade also appear to be independent spillover events from marine mammals (Fig. 2A, B). This is further supported by the fact that these viruses share mutations in PB2 that are associated with mammalian adaptation and are present in viruses forming the marine mammal clade (Fig. 3).

### Lower evolutionary rate of H5N1 HPAI viruses in marine mammals

If H5N1 HPAI viruses are transmitting independently in marine mammals across multiple South American countries, then a host-specific local clock (HSLC) can be used to accommodate a different rate of evolution (Supplementary Fig. 7). Using an HSLC, the estimated rate of evolution in the marine mammal clade (human and avian viruses excluded) was ~ 2-fold lower ($2.5 \times 10^{-3}$; $2.0$–$3.0 \times 10^{-3}$ 95% HPD) than the avian rate ($5.4 \times 10^{-3}$; $4.9$–$5.9 \times 10^{-3}$ 95% HPD), which includes wild birds and poultry but excludes spillovers into mammals (Fig. 2C). The marine mammal rate was still ~ 2-fold lower compared to birds when only the third codon position was considered (Fig. 2C). Importantly, similar tree topologies were inferred using the entire virus genome (~ 13 kb) (Supplementary Fig. 7) and the third codon position only (Supplementary Fig. 8), which excludes any adaptive mutations that would be selected for in marine mammals following a host-switch.

### Global SNP analysis reveals mammal adaptation mutations and suggests two H5N1 HPAI subpopulations during mammal-to-mammal transmission in Argentina

Across the genome, we identified more than 64 amino acid changes in the H5N1 HPAI viruses from Península Valdés when compared with the original Goose/Guangdong (Gs/Gd) strain (Supplementary Table 3). Of the 64 mutations, 14 are potentially associated with increased virulence, transmission, or adaptation to mammalian hosts, and sixteen are present in H5N1 viruses from Argentina's coastal outbreaks in marine mammals and terns but absent in H5N1 (B3.2 genotype) strains from North America and from goose/poultry strains from Argentina (Supplementary Table 4). Of note, eleven of the sixteen common mutations were also present in the human case in Chile.

Argentina's marine mammal viruses inherited eight amino acid changes that emerged previously in marine mammals in Chile and Peru but were never seen in H5N1 HPAI viruses circulating in birds in those countries and appear to be specific to the marine mammal clade (Fig. 3): Q591K and D701N in PB2; L548F in PB1; A20T, M86I, and M548I in PA; and R21Q and I226T in NS1. Almost all mutations (except L548F in PB1) were also present in the two Brazilian and the six Uruguayan marine mammal viruses. The conservation of seven amino acid changes across all marine mammal viruses collected from four countries over eight months (Chile, Argentina, Brazil, Uruguay; March through October) further supports the existence of an independent chain of virus transmission among marine mammals, separate from avian transmission chains. In addition to nonsynonymous mutations, four silent mutations in PB1 (A1167T), PA (C1359T), and NP (C669T and T1239C) were found in marine mammal viruses in Argentina that were inherited from marine mammal viruses circulating in Peru and/or Chile

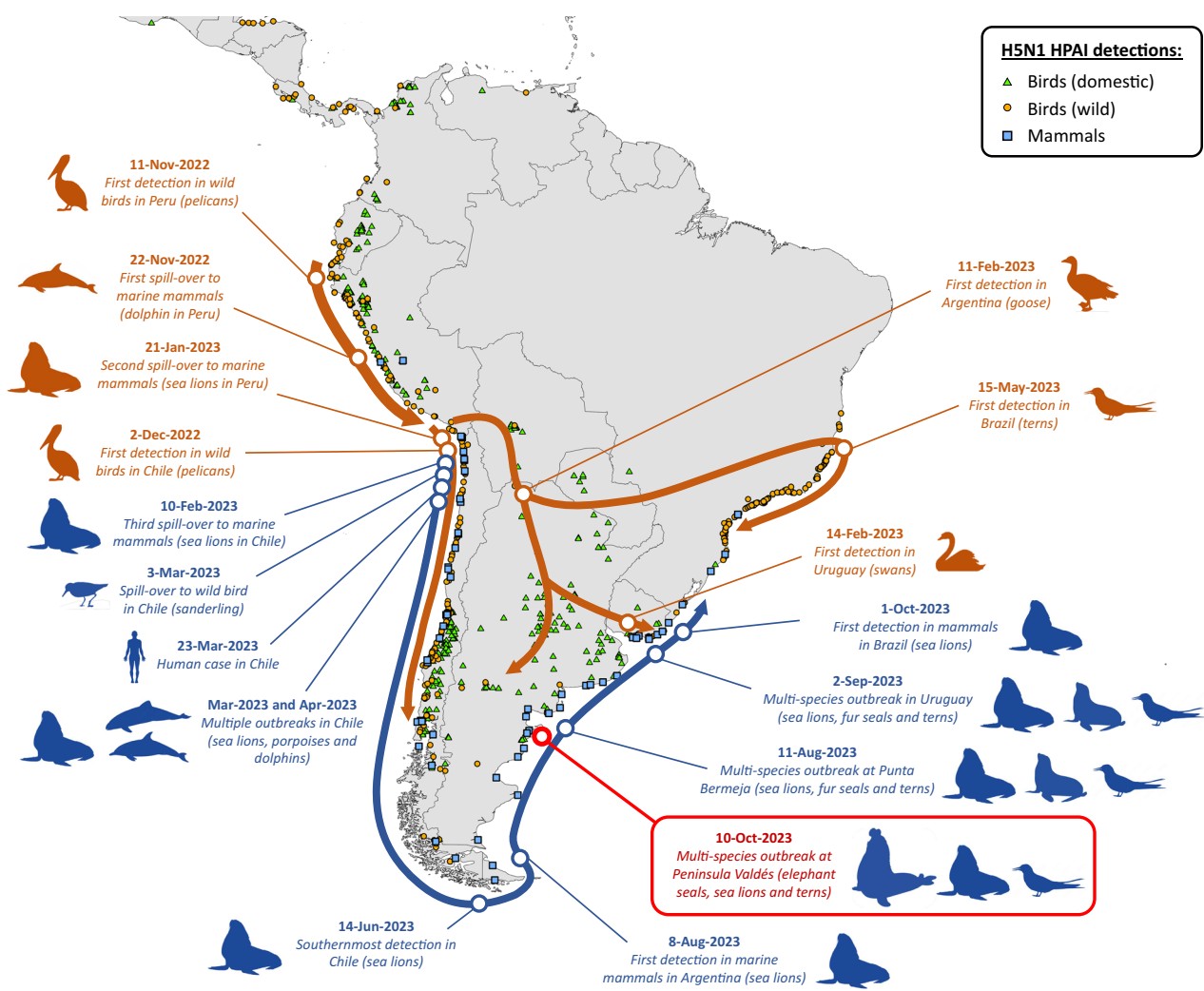

**Fig. 4 | Chronology and hypothesized pathways of spread of H5N1 HPAI (2.3.4.4b) viruses in South America, 2022–2023.** H5Nx HPAI detections (1-Sep-2022 to 31-Dec-2023) reported to the World Animal Health Information System (WAHIS/WOAH) and by the Chilean Servicio Agrícola y Ganadero (SAG) are represented by orange circles (wild birds), green triangles (domestic birds) and blue squares (mammals). Note that there are significant differences in surveillance strategies among countries that may produce gaps or distortions in the geographic distribution of H5Nx HPAI detections and the presumed pathways of virus spread. The location of the outbreak investigated in this study (Península Valdés) is highlighted in red. Arrows represent the timeline of hypothesized pathways of virus spread, as derived from the chronology of detections and our phylodynamic analysis. The pathways of virus spread and significant events of the avian and marine mammal clade viruses are represented in dark orange and dark blue, respectively. Note that virus spread pathways in this figure are intended as a conceptual model and are not geographically precise. Source data for the geographic locations of H5Nx HPAI detections are provided as a Source Data file.

(Supplementary Fig. 9). These mutations were all present in viruses from oronasal, tracheal, lung, brain and rectal swabs of one elephant seal pup (CH-PD032), further corroborating that they were not de novo mutation events.

Synonymous and non-synonymous mutations also occurred during H5N1 2.3.4.4b circulation in the Atlantic (eastern) coast of South America, leading to the evolution of two distinct subpopulations defined by specific mutations. The first subpopulation is defined by a new V122I substitution in PB2 and the loss of the L548F substitution in PB1 (owing to a secondary substitution) and was detected in Argentina, Uruguay and Brazil. In most cases, there was an additional E237A substitution in PA, which was also detected in marine mammals in Brazil and Uruguay (Fig. 3 and Supplementary Table 4). The second subpopulation is defined by a new Q621K substitution in PB1 and was detected exclusively in Argentina (Fig. 3 and Supplementary Table 4), which in many cases is accompanied by mutation A133S in HA (H5 numbering). This A133S substitution in HA that was seen in Argentina in South American terns ($n = 2$), South American sea lions ($n = 4$), and an elephant seal ($n = 1$) (note: this HA region could not be sequenced

from the South American fur seal) was not observed in previous H5N1 marine mammal viruses in South America, North America or Europe, nor in other bird viruses from South America (Fig. 3 and Supplementary Table 4). Of note, both subpopulations were found in the terns and elephant seals sampled for this study and in mammalian and avian hosts in a multi-species outbreak at Punta Bermeja (~ 260 km north of Punta Delgada) in August 2023, but not in the H5N1 viruses that circulated in a wild goose and in poultry in Argentina from February to July 2023 (Fig. 3 and Supplementary Tables 3, 4). Figure 4 summarizes our hypothesized pathway of spread of H5N1 HPAI viruses in South America based on the molecular evidence and the chronology of reported detections.

## Discussion

Since 2020, the world has witnessed a global epizootic of H5N1 clade 2.3.4.4b viruses with substantial ecological impact on wildlife species, including pinnipeds. Although H5N1 HPAI viruses were previously implicated in the mortalities of harbor seals (*Phoca vitulina*) and gray seals (*Halichoerus grypus*) in Europe in 2016–2021[35–37] and in North

America in May–July 2022[24,26], the magnitude of those mortalities (< 200 deaths in total) would pale in comparison with the impacts that ensued when these viruses arrived in South America. At least 30,000 sea lions have died in Peru, Chile, Argentina, Uruguay, and Brazil[11,13–15,27,29–32]. In addition, HPAI caused the largest mortality event of elephant seals recorded to date, with the death of > 17,000 pups and an unknown number of adults at Península Valdés, Argentina[33]. Our epidemiological account of this outbreak provides clinical observations with ecological context for H5N1 HPAI infection in elephant seals. Furthermore, our viral genome data provides evidence for the evolution of a novel marine mammal clade of H5N1 (2.3.4.4b) HPAI virus that has spread among pinnipeds in several countries of South America, revealing mutations that may have enabled their ability to infect mammals while also retaining the ability to spillover to avian hosts.

While serological surveys indicate broad exposure to influenza A viruses (IAV) in pinnipeds globally, mass mortality events have been rare[38–40]. Prior to 2022, the largest IAV outbreak in pinnipeds occurred in 1980, when H7N7 HPAI viruses killed 400–500 harbor seals at Cape Cod, USA, representing ~ 20% of the species' local population[41,42]. Other significant pinniped mortalities attributed to IAV comprise the death of 162 harbor seals in New England, USA, in 2011 due to H3N8 strain[43] and 152 harbor seals in Denmark in 2014 due to H10N7 strain[44]. Prior to 2023, no pinniped deaths had been attributed to IAV in South America. There are also no published studies reporting on the detection of IAV (or antibodies against them) in southern elephant seals. For northern elephant seals (*Mirounga angustirostris*), the only IAV detections were asymptomatic infections with human-like H1N1 strains in California, USA, in 2009–2012 and 2019[45–47]. Considering that IAV surveys on the Atlantic coast of South America have only reported low pathogenicity avian influenza (LPAI) H11 and H13 strains in coastal birds[48–50] and antibodies against H1 strains in fur seals[39], it is likely that southern elephant seals at Península Valdés were naïve to H5 viruses until 2023.

Our data shows that elephant seal pups were severely impacted by H5N1 at Península Valdés, but the extent to which adult elephant seals were affected by HPAI is unclear. The unusually high number of adult carcasses at Punta Delgada, as well as the abnormal haul-outs and the confirmed case in Golfo Nuevo reported here, suggest that adult elephant seals are susceptible to H5N1 (2.3.4.4b) HPAI infection. Furthermore, a dead adult male elephant seal at Punta Tombo (a non-breeding area ~ 190 km southwest of Punta Delgada) was confirmed positive for H5N1 HPAI virus by national authorities in mid-September 2023[51], prior to the onset of the Península Valdés outbreak. Beyond recorded beached carcasses, the complete disruption of the social and breeding structure at Punta Delgada (evidenced by the absence of harems and large alpha males and the presence of motherless pups) suggests that adult elephant seals abandoned the colony prematurely, perhaps after becoming infected. Yet, it is difficult to ascertain the number of adult deaths, which may have happened at sea and will only be accounted for via population censuses at Península Valdés in coming years. Nevertheless, the fact that in 2023 the adult females abandoned the beach probably before being impregnated (which normally occurs when pups are weaned[52]) suggests that this population will likely experience an atypically low birth rate in 2024, even if most adult females survived.

From a disease evolution standpoint, there is growing concern that H5N1 viruses adapted to mammalian transmission could facilitate host jumps to other species, including humans. Mammal-to-mammal IAV transmission is believed to have occurred sporadically among pinnipeds over the years[41–44,53,54]. The recent demonstration that the H5N1 strain from a human case in Chile (which belongs to the marine mammal clade discussed in this study) is transmissible between co-housed ferrets[55] also supports the notion that mammal-to-mammal transmission could have played a role in the spread of these viruses in marine mammal communities in South America. We posit that the high

mortality rate in elephant seal pups is also consistent with mammal-to-mammal transmission, as pups are toothless and nurtured exclusively through nursing from their mothers. Contact with wild birds is minimal and could not explain the death of ~ 95% of all pups born (~ 17,000) in a matter of weeks, over 200 km of coastline along Península Valdés. Some newborns may have been infected before birth, as transplacental transmission of H5N1 HPAI viruses has been reported in humans[56], and high virus loads were detected in aborted sea lion fetuses[11,30]. It could also be that mothers were infected and shed the virus through their milk, infecting their pups[57,58]. Yet, how adult female elephant seals would have been infected in the first place without mammal-to-mammal transmission presents a thornier question. Prior to arriving to give birth at Peninsula Valdés in September, the elephant seals would have spent a solitary winter at sea in the South Atlantic and Southern Oceans[59]. Feeding is an unlikely route of exposure to H5N1 HPAI viruses since elephant seals do not eat birds or mammals, feeding instead on squid, fish, and crustaceans captured in deep waters[52,60], and adult elephant seals will fast while on land[52,61]. Moreover, elephant seals are pelagic and only come to shore and aggregate for a few weeks to breed and later to molt, thus limiting the time window for interspecific interactions and transmission on land[52,62]. The main interactions between birds and elephant seals involve opportunistic scavenging of elephant seals' placental remains, molted skin, and carcasses by gulls[61] (Supplementary Fig. 1E), which provides more opportunities for mammal-to-bird transmission than vice-versa. In this context, it seems unlikely that bird-to-mammal transmission alone could explain the simultaneity, extent, and speed of the outbreak in elephant seals at Península Valdés. Although there are still many unknowns about the precise viral transmission routes (e.g., contact, environmental, aerosol), seal-to-seal transmission seems the most plausible hypothesis to explain viral dissemination during this outbreak. In experimental infections in several wild mammals and in ferret models, nasal and oral H5N1 virus shedding has occurred as quickly as one-day post-inoculation and lasted about a week[63], suggesting that elephant seals infected soon after hauling out on the beach could have begun shedding virus within a very short period of time.

How the virus was first introduced to the elephant seals when they arrived at the beaches is unclear, but sea lions appear to be the most likely source. Notably, the epidemic path of HPAI along coastal Argentina left virtually no rookery or stretch of beach without dead or symptomatic sea lions from south to north[30,64] and then progressed to neighboring Uruguay and Brazil[31,32]. This extended spread along the Atlantic coast mirrored that seen along the Pacific coast, with the common denominator being infected sea lions[11,27] (Fig. 4). South American sea lions regularly visit multiple rookeries and haul-outs, sometimes interacting aggressively with other pinnipeds[65,66]; furthermore, several beaches of Península Valdés are shared by sea lions and elephant seals breeding in close proximity (Supplementary Fig. 10). At Punta Delgada, we observed numerous sea lion carcasses amongst dead elephant seals (Fig. 1A and Table 2) and witnessed aggressive interactions between sea lions and elephant seals (Supplementary Fig. 1C, D). Government veterinarians who monitored sea lion rookeries in Argentina noted that some animals showing clinical signs of HPAI survived for several days and often abandoned the rookeries while ill (Veronica Sierra, pers. comm.). It is plausible that these sea lions visited different sites during their convalescent period, including elephant seal colonies, and may have played a key role in the spread of H5N1 viruses. Considering the occurrence of a multi-species outbreak with sea lions, fur seals, and seabirds one month prior in the region[30], it is conceivable that birds infected in that outbreak also contributed to the spread of the virus to the beaches occupied by elephant seals in Península Valdés. In addition, mammal-to-bird spillovers do not seem improbable given the frequent observations of gulls and other avian scavengers feeding on sea lion and elephant seal carcasses in Argentina (Supplementary Fig. 1E). It is interesting to note that the earliest H5N1

HPAI detection in the Falkland/Malvinas Islands was that of a marine mammal clade virus in a southern fulmar (*Fulmarus glacialoides*) in late October 2023[67]. Fulmars are known to occasionally scavenge on birds and mammals[68–70], and this detection raises the possibility that scavenging procellariiform birds could also play a role in the spread of these viruses. On the other hand, it is still unclear how terns (which are not scavengers) became infected, and further studies may help to clarify whether other birds (e.g., gulls) played a role as bridging hosts between pinnipeds and terns.

Mammal-to-mammal transmission is also supported by our regional phylogenetic analysis, which identified a novel H5N1 2.3.4.4b clade with viruses that appear to be specific to marine mammals. This marine mammal clade comprises strains with mutations that were not present in H5N1 2.3.4.4b viruses in birds (wild and domestic) from Peru, Chile, Argentina, Uruguay, and Brazil, excepting occasional spillovers from marine mammals to coastal birds (terns and sanderling; Supplementary Tables 3, 4). Some of these mutations (such as Q591K and D701N in PB2) are associated with increased virulence, transmission, or adaptation to mammalian hosts[71–73] and have been maintained since they first emerged in H5N1 HPAI viruses in marine mammals in Chile. The maintenance of a unique cassette of mutations in viruses from marine mammals (Fig. 3), the lower rate of evolution of these viruses (Fig. 2C), and the distinct pathways of spread across host groups (Fig. 2B) and geographical areas (Fig. 4), strongly support the hypothesis that viruses from the novel H5N1 marine mammal clade had an independent chain of virus transmission among marine mammals, separate from the avian transmission chains in Argentina and other countries, and retained the capacity to spillover to terns. Of note, the rate of evolution of the marine mammal viruses is not only lower than that of H5N1 viruses circulating in birds in South America, but it is also considerably lower than that of H5N1 viruses circulating in cattle in the USA ($6.2 \times 10^{-3}$; $5.3–7.2 \times 10^{-3}$ 95% HPD)[74]. This could relate to the high densities of susceptible hosts in farmed animals (dairy cows or poultry) and colonial seabirds providing opportunities for rapid and sustained transmission. To date, no reassortment has been observed between H5N1 HPAI viruses in South America and LPAI viruses belonging to the South America lineage that circulate enzootically in aquatic birds in Argentina, Chile, and Peru[11,75].

To our knowledge, the H5N1 HPAI 2.3.4.4b marine mammal clade identified in South America represents multinational transmission of HPAI in mammals, an occurrence not previously reported. Over the last century, LPAI H1, H2, H3, and H7 viruses have periodically jumped into mammals, including humans, swine, canines, and equines, causing major outbreaks and pandemics[76]. Spillover of H5N1 2.3.4.4b clade also regularly occurs in humans and terrestrial and marine mammals on a global scale, but onward transmission in mammals is limited and not sustained over time, leading to speculation that the H5 subtype perhaps is not capable of causing a pandemic[26,77,78]. Despite gaps in the available data, our epidemiological and phylogenetic results support the hypothesis that the spread of viruses from the novel marine mammal clade in South America has occurred via mammal-to-mammal transmission. While there is a need for a better understanding of the mode of transmission between marine mammals, there is increasing consensus that mammal-to-mammal transmission has played a significant role in the recent spread of H5N1 HPAI viruses worldwide[79]. The recovery of live viruses for pathogenesis and transmission studies would be valuable to demonstrate how these strains behave in mammalian experimental models. Furthermore, additional studies on the virus prevalence, shedding, and genome in different potential hosts within the coastal wildlife of Patagonia, especially for species that were already shown to be susceptible to H5N1 HPAI viruses in other regions (e.g., skuas, gulls, petrels) would be helpful to identify if there are asymptomatic reservoirs of infection and bridge hosts.

The implications of sustained mammal-to-mammal transmission of H5N1 HPAI viruses could be far-reaching, both from a conservation and a public health perspective. From the standpoint of wildlife conservation, this is particularly concerning for endangered pinnipeds with limited geographic distribution such as Caspian seals (*Pusa caspica*) and Hawaiian monk seals (*Neomonachus schauinslandi*), among others[80]. Significant mortalities of southern elephant seals and Antarctic fur seals (*Arctocephalus gazella*) have been attributed to H5N1 HPAI in South Georgia[67,81]; however the viruses involved do not belong to the marine mammal clade identified in this study, clustering instead with avian viruses from inland Argentina[67]. Considering that 95% of Antarctic fur seals[82] and 50% of southern elephant seals[83] breed in South Georgia, these populations could be at great risk if the marine mammal clade viruses spread there in the future. The detection of marine mammal clade viruses in dead dolphins and porpoises in Chile[13,14] is also concerning since 23% of the world's odontocete species are already threatened with extinction[84]. If pinnipeds become a sustainable reservoir for H5N1 HPAI viruses that retain the capacity to infect wild birds, coastal bird species could be repeatedly affected by spillover infections. Furthermore, the implications could become even more severe if the marine mammal clade viruses evolve to enable transmission among terrestrial mammals or if additional gene reassortment occurs with South American LPAI viruses present in Argentina[48,50,85,86], potentially expanding either the host range, pathogenesis, and/or transmission in wildlife.

From a public health perspective, mammal-to-mammal transmission could be a stepping-stone in the evolutionary pathway for these viruses to become capable of human-to-human transmission[87]. As mentioned previously, some of the mutations found in the strains of the marine mammal clade are already known to be of concern. In particular, the mutation D701N in PB2 has been shown to compensate for the lack of the E627K mutation in PB2 in terms of improved viral growth in mammalian cells and enhanced aerosol transmission of H3N2 and H5N1 viruses[88]. The fact that the H5N1 HPAI virus detected in a human case in Chile belongs to the marine mammal clade and is transmissible among ferrets[55,89] highlights the potential risk to public health. In addition, the possibility of zoonotic strains resulting from the reassortment between HPAI viruses and other IAV strains infecting pinnipeds is also of concern[43,90], especially since pinnipeds are known to occasionally host human-like IAV strains[45–47]. However, the phenotypic effects of mutations in other gene segments found in the H5 viruses from our study (Supplementary Table 4) are not yet known, and further research using in vivo mammalian models is needed to determine whether they can enhance virulence and/or transmission.

In conclusion, the world has seen a concerning increase in the number of H5N1 HPAI detections in mammals since 2023, including notable outbreaks such as the one reported here. Amidst growing evidence that mammal-to-mammal transmission played a role in H5N1 HPAI outbreaks in dairy cows in North America[57,91] and in fur farms in Europe[92,93], the outbreak among elephant seals in Península Valdés represents another case where mammal-to-mammal transmission was potentially involved in the spread of H5N1 HPAI infections, this time in free-ranging wildlife. Genetic drift and shift in IAVs are stochastically-driven phenomena[94], and mutations that increase transmissibility between mammals are more likely to occur in mammals than birds[95,96]. Therefore, the recent increase in H5N1 HPAI circulation in mammals is a warning that should not be ignored. Moving forward, HPAI management requires holistic strategies that recognize the interconnectedness of human, animal, and environmental health as well as safeguard biodiversity, promote sustainable practices, and enhance resilience globally to emerging infectious diseases.

## Methods
### Study species
Southern elephant seals are widely distributed in Subantarctic islands, with a single continental colony at Península Valdés, Patagonia,

Argentina (representing ~ 5% of the global population)[80]. The species has a well-defined annual life cycle, which we summarize as follows based on studies at Península Valdés[52,62]. Adult (and subadult) males and females haul out in late August and early September, with alpha males establishing and defending harems (median 11–13 females per harem, with a maximum of 134 females); subordinate males are chased away but remain along the margins of harems. The number of adults on the beach rapidly increases during the second half of September, reaching its peak by the end of that month. Most females are pregnant when they come ashore, giving birth within $5.7 \pm 1.9$ days after arrival (80% of pups are born by 2 October). Pups are toothless and will nurse for $22.4 \pm 1.7$ days; during this period, the females will fast and remain with their pups, under the protection of the alpha male. Copulations will begin $20.3 \pm 2.1$ days after parturition, i.e., shortly before females wean their pups. The female then abandons the pup and returns to the sea to forage; on average, females spend a total of $28.2 \pm 2.5$ days ashore, fasting. Males also fast on land and will abandon the beach approximately at the same time as females; adult seals are nearly absent on the beach by mid-November. The number of weaned pups will increase during the second half of October, reaching its peak by the end of November. Weaners will remain on the beach for > 5 weeks, fasting while they complete their physiological development and are ready to go to sea to forage. Juveniles and adults will return to the beaches later in the season to undergo molt, with juveniles molting earlier (November to January) than subadults and adults (December to February).

## Study site and field observations
Península Valdés is located in Chubut, Argentina, and is a UNESCO World Heritage site of global significance for the conservation of marine wildlife. We studied two sites at Península Valdés: the elephant seal breeding colony at Punta Delgada and the interior beaches of Golfo Nuevo, where sporadic seal haul-outs occur. Punta Delgada (from 42.753°S 63.632°W to 42.771°S 63.649°W) is a 3 km beach on the exposed seashore of Península Valdés (Supplementary Fig. 1) where southern elephant seals breed in high densities[97,98]. Field surveys were conducted on 4-5-Oct in 2013, 2015 and 2022 (baseline years), and during the mortality event on three occasions, 10-Oct-2023, 3-Nov-2023 and 13-Nov-2023. In each survey, a team equipped with binoculars walked along the clifftop to count live and dead elephant seals, differentiating individuals by sex and age class (pup, weaner, juvenile, subadult male class 1–4, adult male, adult female) and male dominance status (alpha or subordinate)[99,100]. Because pups and weaners have extremely limited mobility and cannot leave the beach for > 7 weeks until they have finished their development, pup survival in 2023 was estimated by dividing the number of living pups/weaners counted on 13-Nov-2023 by the total number of pups/weaners counted on 10-Oct-2023. For outbreak investigation in 2023, a second team of trained veterinarians wearing full PPE descended to the beach to document clinical signs, collect samples from affected animals, and count the carcasses of other wildlife species. We also covered a 50-km stretch of interior beach in Golfo Nuevo, from Cerro Prismático (42.595°S 64.811°W) to Cerro Avanzado (42.835°S 64.874°W), including the city of Puerto Madryn (~ 130,000 inhabitants) (Supplementary Fig. 2). Elephant seals do not breed in this area, but sporadic haul-outs are reported by the public and park rangers to the Red de Fauna Costera de la Provincia del Chubut (RFC). Data on the age, sex, condition, location, and date of each seal were extracted from RFC records for 2022 and 2023.

## Sample collection
On 10-Oct-2023, a team of trained veterinarians wearing full PPE descended to the beach at Punta Delgada to collect samples from affected animals. Post-mortem swabs (oronasal, rectal, tracheal, lung, and brain) were collected from four elephant seal pups, six South American terns (*Sterna hirundinacea*), and two royal terns (*Thalasseus maximus*) found dead (carcasses still in *rigor mortis*). On 1-Nov-2023, swabs were obtained from a subadult male elephant seal that hauled out and died in Golfo Nuevo. Swabs were placed in cryotubes containing 1 mL of DNA/RNA Shield (#R1100-250, Zymo Research, Irvine, CA, USA) for inactivation, and stored in a cooler with icepacks, then transferred to − 80 °C within 24 h.

## Virus detection
Samples from four elephant seal pups, five adult South American terns, and two adult royal terns were pooled according to species and sample type. An additional pool containing all samples from a juvenile South American tern (brain, lung, oronasal, and rectal) from Punta Delgada and a separate pool containing all samples from a dead subadult male elephant seal (oronasal, rectal, and lung) from Golfo Nuevo were also analyzed. Viral RNA was extracted from 140 μL of suspension from swabs using a QIAamp Viral RNA Mini Kit (#52904, Qiagen, Valencia, CA, USA). RNA was eluted in a final volume of 60 μL and stored at − 80 °C. Viral cDNA was prepared using 15 μL of viral RNA and random hexamers in a final volume of 30 μL using a High-Capacity cDNA Archive kit (#4368813, Applied Biosystems, Foster City, CA, USA). The cDNA from all pooled samples were tested for influenza A viruses by RT-qPCR using TaqMan Universal PCR Master Mix (#4304437, Applied Biosystems) directed to the matrix gene (forward: 5′-GAC CRA TCC TGT CAC CTC TGA-3′, reverse: 5′-AGG GCA TTY TGG ACA AAK CGT CTA-3′, probe: 5′-FAM-TGC AGT CCT CGC TCA CTG GGC ACG-TAMSp-3′)[101]. Positive samples from elephant seals were then tested using primers and probes for H5 clade 2.3.4.4b detection (forward: 5′-CCT TGC GAC TGG GCT CAG-3′, reverse: 5′-ATC AAC CAT TCC CTG CCA-3′, probe: 5′-FAM-AGA AGA AAR AGA GGG CTG TTT GGG GCT-BHQ-1-3′)[102]. Quantification cycle (Cq) values were used as a proxy to compare viral RNA load in different samples and to facilitate sample selection for full genome sequencing. RT-qPCR reactions were performed on an ABI Prism 7500 SDS (Applied Biosystems).

## Full genome sequencing
The viral genome was amplified from RNA using a multi-segment one-step RT-PCR with Superscript III high-fidelity RT-PCR kit (#12574035, Invitrogen, Carlsbad CA) according to manufacturer's instructions using the Opti1 primer set (Opti1-F1: 5′-GTT ACG CGC CAG CAA AAG CAG G-3′, Opti1-F2: 5′-GTT ACG CGC CAG CGA AAG CAG G-3′, Opti1-R1: 5′-GTT ACG CGC CAG TAG AAA CAA GG-3′)[103]. Amplicons were visualized on a 1% agarose gel and purified with Agencourt AMPure XP beads (#A63881, Beckman Colter, Brea, CA). The concentration of purified amplicons was quantified using the Qubit High Sensitivity dsDNA kit (#Q32850, Invitrogen) and a Qubit Fluorometer (Invitrogen). The sequencing library was prepared with the Rapid Barcode library kit (#SQH-RBK110.96, Oxford Nanopore, Oxford, UK) and loaded on the Mk1c sequencer according to ONT instructions for the R.9 flow cells. Real-time base calling was performed with MinIT (Oxford Nanopore); the automatic real-time division into passed and failed reads was used as a quality check, excluding reads with a quality score < 7. Quality-checked reads were demultiplexed and trimmed for adapters and primers, followed by mappings and a final consensus production with CLC Genomics Workbench v23.0.2 (Qiagen).

## Phylogenetic analysis
To place the coastal Argentinean viruses in a global context, we downloaded HA gene sequences from H5N1 HPAI clade 2.3.4.4b viruses globally from GenBank and GISAID since January 1, 2021. Phylogenetic relationships were inferred for the HA gene using the Maximum likelihood (ML) methods available in IQ-Tree 2[104] with a GTR model of nucleotide substitution with gamma-distributed rate variation among sites. Due to the size of the dataset, we used the high-performance computational capabilities of the Biowulf Linux cluster at

the National Institutes of Health (http://biowulf.nih.gov). To assess the robustness of each node, a bootstrap resampling process was performed with 1000 replicates.

To study how the H5N1 HPAI outbreaks in Argentina were connected to outbreaks occurring in other South American countries, we performed a phylogenetic analysis of 18 available H5N1 virus genomes from three species of marine mammals and two species of terns in coastal Argentina, along with 249 closely related H5N1 virus genomes obtained from avian and mammalian hosts in five South American countries (Argentina, Brazil, Chile, Peru, Uruguay) and South Atlantic islands (Falkland/Malvinas and South Georgia) available from GISAID and/or GenBank public databases (Supplementary File 2). Alignments were generated for each of the eight segments of the virus genome (PB2, PB1, PA, HA, NP, NA, MP, and NS) using MAFFT v7.490[105]. Phylogenetic trees were inferred for each segment individually using maximum-likelihood methods with a GTR + G model of nucleotide substitution and 500 bootstrap replicates using the CLC Genomics Workbench v23.0.2 (Qiagen), and the inferred trees were visualized. Since the H5N1 viruses were collected from a recent outbreak and had little time to accrue mutations and diversify, limiting genetic diversity, all Bayesian analyses were performed using concatenated genome sequences (13,140 nt) to improve phylogenetic resolution (after removing reassortants and viruses that did not have sequence data available for all eight segments).

We performed a time-scaled Bayesian analysis using the Markov chain Monte Carlo (MCMC) method available using the BEAST package pre-release v1.10.5 (compiled on 24-Apr-2023)[106], using GPUs available from the NIH Biowulf Linux cluster (http://biowulf.nih.gov/). First, the analysis was run with an exponential growth demographic model, a GTR + G model of nucleotide substitution, and an uncorrelated lognormal relaxed molecular clock. To account for the possibility that high rates of convergent evolution involving adaptive mutations following host switches (see mutation analysis below) could artificially cluster marine mammal viruses on the tree that do not actually share a common ancestry, a second tree was inferred for the third codon position only. The MCMC chain was run separately 3–5 times for each dataset using the BEAGLE 3 library[107] to improve computational performance, until all parameters reached convergence, as assessed visually using Tracer version 1.7.2[108]. At least 10% of the chain was removed as burn-in and runs for the same dataset were combined using LogCombiner v1.10.4[106]. An MCC tree was summarized using TreeAnnotator v1.10.4[106].

After the initial analysis determined that the vast majority of H5N1 viruses collected from marine mammals clustered together in a well-supported clade (posterior probability = 1.0), in both the whole genome and third codon analyses, we repeated the BEAST analysis using a host-specific local clock (HSLC)[109] to accommodate differences in the evolutionary rate between marine mammals and avian hosts. For the HSLC analysis, any singleton avian and human viruses positioned in the marine mammal clade (likely representing transient dead-end spillovers) were excluded to ensure monophyly (the four viruses detected in terns in this study were also excluded). Similarly, any singleton marine mammal viruses positioned in the major avian clade (which also likely represent transient dead-end spillovers from birds to marine mammals) were excluded.

To compare evolutionary rates in marine mammals and avian hosts across the eight different segments of the virus genome, the analysis was repeated using eight genome partitions (PB2, PB1, PA, HA, NP, NA, MP, NS). A phylogeographic discrete trait analysis[110] was performed to quantify rates of viral gene flow between different host groups (wild bird, poultry, marine mammal, terrestrial mammal (which includes humans and a zoo lion)) as well as between locations (Argentina, Brazil, Peru, Chile, Uruguay, South Atlantic). Since extensive virus gene flow was observed between Chile/Peru, which is not the focus of this study, a single combined Chile/Peru

location category was used. A location state was specified for each viral sequence based on the host species and the location of the collection. The expected number of location state transitions in the ancestral history conditional on the data observed at the tree tips was estimated using "Markov jump" counts[111,112], which provide a quantitative measure of asymmetry in gene flow between defined populations.

## Mutation analysis
Consensus nucleotide sequences for the eight open reading frames were translated to protein and compared to viruses from birds and mammals from Argentina, other South American countries, Antarctica, North America (genotype B3.2 from 2022–2023), and reference strains from Asia (A/goose/Guangdong/1/1996 and A/Vietnam/1203/2004).

## Reporting summary
Further information on research design is available in the Nature Portfolio Reporting Summary linked to this article.

## Data availability
The sequence data generated in this study have been deposited in GenBank under accession codes PQ002111–PQ002158 and PP488310–PP488349. The extensible markup language (XML), maximum clade credibility (MCC), and maximum likelihood (ML) trees, host-specific local clock (HSLC) model files, Markov jump analysis files, GISAID acknowledgement tables, and underlying data for raw tree files are provided in Zenodo [https://doi.org/10.5281/zenodo.13923371]. Source data are provided in this paper.

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

## Acknowledgements

Wildlife Conservation Society, University of California, Davis, and the National Institute of Agricultural Technology (INTA) (PNSA PD114) funded this study. This work was also partially supported by Alexander von Humboldt Foundation, through a fellowship to A. Rimondi. Special thanks go to M. Muñoz and A. Puebla from Unidad Genómica (INTA) for their outstanding technical support in sequencing IAV genomes. We thank Lic. M. Cabrera from Dirección de Conservación, Secretaría de Turismo Municipal de Puerto Madryn, and park rangers from Área Natural Protegida El Doradillo. We also thank H. O. Loza from Reserva Natural de la Defensa Faro Punta Delgada (Armada de la República Argentina) and the navy personnel who work there. We acknowledge data shared by Red de Fauna Costera de la Provincia de Chubut. Permits were granted by Dirección de Fauna y Flora Silvestres de la Provincia de Chubut and Subsecretaría de Conservación y Áreas Protegidas de Chubut. We acknowledge Servicio Nacional de Sanidad y Calidad Agroalimentaria (SENASA) and other laboratories in South America that submitted H5N1 virus sequence data to GISAID.

## Author contributions

Author contributions were as follows: Study design: M.U. and A.R. Funding: M.U., V.F., and A.R. Sample and data collection: M.U., R.E.T.V., J.C., V.Z., V.F., and C.C. Virus detection and virus sequencing: V.S.O. and A.R. Phylogenetic analyses: M.I.N., A.R., and P.L. Data analysis and interpretation: M.U., R.E.T.V, M.I.N., and A.R. Writing of the manuscript: M.U., R.E.T.V., M.I.N., and A.R. All authors approved the manuscript before submission.

## Competing interests

The authors declare no competing interests.
