## [Peer Review file · Nature Communications]

Massive outbreak of influenza A/H5N1 in elephant seals in Argentina supports mammal-to-mammal transmission

Corresponding Author: Dr Marcela Uhart

Editorial Note: Parts of this peer review file have been redacted as indicated to avoid any copy right infringement.

Version 0:

Reviewer comments:

Reviewer #1

(Remarks to the Author)

Uhart and colleagues present a detailed study on the large-scale outbreak of the H5N1 virus in elephant seals at Península Valdés, Argentina, a UNESCO World Heritage site for marine wildlife conservation. The authors provide epidemiological data and full genome characterization of H5N1 clade 2.3.4.4b viruses linked to the outbreak in elephant seals and concurrent tern mortality. They analyze data from this event and prior reports to investigate potential pathways of H5N1 virus transmission among marine mammals and birds in South America and document a rapidly spreading H5N1 marine mammal clade carrying mammalian adaptation mutations of potential public health concern. The dataset is valuable for understanding the mammalian spillover of a pathogen with pandemic potential and migratory patterns across South America. However, there are critical issues regarding the clarity of the timeline, robustness of the conclusions, and adequacy of data sampling.

Major

1. The timeline of events and data collection is unclear. For example, the statement about the peak mortality of elephant seal pups between 25 September 2023 and 10 October 2023 conflicts with an earlier sentence indicating that the survey was not conducted until 10 October 2023.

2. The study's modest number of viral genomes and limited sampling from wild birds raise questions about the robustness of their conclusions, particularly regarding primary transmission routes. The hypothesis that transmission occurred primarily through mammalian hosts forming a 'mammalian clade' across vast distances requires more support. A brief overview of the standard movement patterns of elephant seals and their interactions with other potential hosts would help. The high mortality rate in mammals suggests a potential alternate host (apparently healthy aquatic birds?). While the results are presented with certainty, the discussion raises doubts about the hypothesis.

Minor

Introduction: Line 37 needs rephrasing to reflect the likelihood of underreported widespread outbreaks in other regions: "In Europe and southern Africa, impacts on wildlife were particularly severe in seabird colonies, with losses in the tens of thousands." For further context, please see Xie et al., Nature 2023.

Results

* The first section states that signs of unusual mortality were observed around 25-Sep-2023, but the survey was not conducted until 10-Oct-2023. This gap should be addressed to clarify the timeline of events. It mentions mortality peaks and subsequent observations without specifying the methods or frequency of these observations between 25-Sep-2023 and 13-Nov-2023.

* The comparison of mortality rates between 2022 and 2023 lacks detail on how these rates were calculated and whether other factors could have influenced the observed increase. The section notes a 95% mortality rate by 13-Nov-2023 but does not provide information on how these figures were determined, such as the initial population size and the methodology used for monitoring mortality over time.

* Figure 5 is well-constructed and informative. However, the accompanying text should emphasize the limitations of the study, including intermittent surveillance and modest sample size.

Reviewer #2

(Remarks to the Author)

In this paper, Uhart and coauthors describe a massive outbreak of avian influenza in elephant seals and put forward strong evidence for mammal-to-mammal transmission of influenza viruses. The authors observed many dead southern elephant seal pups at Península Valdés compared to the previous year. The authors also observed a breakdown in the social and breeding structure with a lack of adult seals. Pups presented with symptoms that suggested avian influenza. The lack of feeding when out of the water and the limited possible interactions between pups and birds strongly suggested mammal-to-mammal transmission. The authors also noted that many terns also died and showed symptoms of influenza. The peak of tern mortality occurred several weeks after the peak of seal mortality. Sequencing was performed from 1 pup, 1 subadult male that hauled out at an aberrant location after the outbreak at Península Valdés and 3 terns. The authors show that their sequences and other mammalian sequences from Argentina/Chile/Brazil form a monophyletic clade which is separate from the wild bird sequences from inland Argentina. They show that there were 3 jumps from birds into mammals though only one of these jumps led to sustained spread with multiple recorded sequences. The marine mammal clade contains known mutations associated with mammalian adaptation which have been maintained in the terns. The authors suggest that influenza spread via mammal-to-mammal transmission down the west coast of South America to the tip of Chile before spreading up the East Coast of South America through Argentina, Uruguay and Brazil. Finally, the authors claim that there was a lower evolutionary rate in marine mammals compared to the avian virus with all segments having strong purifying selection.

The paper is clearly written and strongly suggests mammal-to-mammal transmission. Figure 5 is an especially well made and useful figure. However, the paper would have benefited from more data to confirm mammal-to-mammal transmission.

The paper presents lots of circumstantial evidence of a substantial avian-flu outbreak in the seals but unfortunately there was only one actual new sequence gathered from a seal pup. Despite the suggestion of many adult deaths, no sequences were gathered from adult seals and the authors suggested that most deaths would happen at sea. There are many unanswered questions about this outbreak. How did the first seal get infected? The authors suggest that it was unlikely through diet as seals eat squid, fish and crustaceans and therefore were not consuming animals infected with influenza. If the seals were infected by sea lions, how was the first seal infected if sea lions interacted more only following the breakdown of social structure? Also, why were the seals infected in October when the virus past through the area in sea lions several months before? Were the infections only seen when the seals hauled out? The roles of sea birds in transmission is also extremely unclear. How did terns get infected? Were other seabirds involved in transmission to or from the seals? The lack of contemporaneous sequencing of birds hinders the ability to really understand the dynamics of influenza in this ecosystem. Is there a possibility that the first seals were infected from environmental contamination from sea birds?

I think that this paper has the potential to be extremely important in demonstrating that there is sustained mammalian spread of H5. The natural history information of the seals very strongly suggests this and I am convinced there is mammal-mammal spread. However, it would be extremely helpful if the authors manage to acquire some more data to bolster their conclusions perhaps by combining data with the group working in South Georgia.

Major Concerns

1. I am concerned by the dN/dS data presented in Supplemental figure 7. The legend does not clearly state what the box and whisker plots shows or what n is. It seems odd to me that all the segments have such identical values. You might expect that there might be positive selection on some segments or that the polymerase segments are less tolerant of mutations. The data is referred to as <0.3 on L178 when it is clearly <0.2.
2. Figure 3 only analyses South American avian sequences but there is no reason not to include global sequences which might be more intensively sampled. Is it possible that reassortment has affected the results of this or other analyses in any way? This should be commented on. The global results for each segment could be included here as a comparison to the South American concatenated tree which would ensure that reassortment had not affected the results.
3. I would ideally like more data to confirm that there is mammal-mammal transmission but if not the discussion should make clear the limitations of the data and what data needs to be collected to confirm the hypothesis.

Minor Concerns

4. In the introduction, please add some focus on mammal-mammal transmission.
5. L92 a graph showing mortality of terns would be nice.
6. L110 exactly how the samples were pooled is not clear here and elsewhere. Please make sure it explicitly states what was pooled and what was sequenced.
7. The phylogeny in figure 2 should include how the clades are related to each other. They should not be presented as separate clades.
8. Figure 3a with only a single human and sanderling case there is no need to present a value. Other values should be presented with a standard error.
9. L170 fix 12.
10. L183 add 'compared to'.
11. L290-291 is there any evidence for this?
12. L304 please comment on why there might be a lower rate of evolution. Perhaps a comparison to work on cows would be appropriate here.
13. L311 jumped.
14. L340-350 please tone this down as there is little risk currently shown to public health.
15. L351-2 please tone this down as many compartments and species remain unaffected by flu.
16. L428 how was this model chosen
17. Is it possible to add extra years of baseline to Table 1

18. Supplementary Figure 3 is not helpful without labels and should be combined with another figure with labels if used.
19. Please be explicit when comparing South American mammals to birds whether terns falling in the mammalian clade and mammals falling in the avian clade have been excluded.

Version 1:

Reviewer comments:

Reviewer #1

(Remarks to the Author)

Reviewer #2

(Remarks to the Author)

The authors have mostly addressed the comments.

The paper itself makes two significant claims – 1. Mass mortality of seal pups due to avian influenza. 2. Mammal-mammal transmission of avian influenza. I believe both of these claims to be true but the authors must be very careful in their claims as the evidence they present is mostly circumstantial. For claim 1, the sequencing evidence confirms that 3 pups had influenza A and given the timing of the virus and symptoms, it is clearly reasonable to assume this is what caused the deaths (even if no healthy individuals were swabbed to ensure they did not have the virus and only a few pups were confirmed to have flu.) For claim 2, the evidence is more circumstantial. There is clearly a large mammal clade, and the biology of the elephant seals means that they are unlikely to have acquired the virus except from conspecifics. However, questions on transmission still remain unanswered. It is possible that a sea lion infected the first seal but there is no particular evidence to support this. We do have evidence of birds being infected with mammalian-adapted sequences and we do not have contemporaneous bird sequence data from near the seals except for the terns which show that large scale infection of birds with mammalian adapted virus is possible. It is therefore possible that the seals could have been infected by an environmental or avian source. I think this is unlikely but it is clearly possible in some cases e.g. in the Falkland islands. The continental spread of the mammalian adapted sequence could easily have been facilitated by birds feeding on corpses leading to a cycle of transmission between birds and pinnipeds. In L352, please be more explicit about the cause of uncertainty rather than saying but as with any hypothesis, this is subject to revision as more data become available. What are the other possibilities and what data would you like?

Please also address the following minor comments.

1. Please add back in that which HA numbering you are using when referring to A133S.
2. It is still unclear to me how and when you pooled samples. Pooling according to sample type is not clear to me. Please make it clear whether you pooled all the samples from a single individual to test for influenza or across multiple individuals. Were any pooled samples then separated for sequencing?
3. L256 please remove “clearly”.
4. L286. I would find it very helpful if the authors could give some information on the timeline of infections that they are envisioning given what is known about incubation time of the virus. This would help demonstrate that most infections likely occurred after haul-out.

REVIEWER #1

Uhart and colleagues present a detailed study on the large-scale outbreak of the H5N1 virus in elephant seals at Península Valdés, Argentina, a UNESCO World Heritage site for marine wildlife conservation. The authors provide epidemiological data and full genome characterization of H5N1 clade 2.3.4.4b viruses linked to the outbreak in elephant seals and concurrent tern mortality. They analyze data from this event and prior reports to investigate potential pathways of H5N1 virus transmission among marine mammals and birds in South America and document a rapidly spreading H5N1 marine mammal clade carrying mammalian adaptation mutations of potential public health concern. The dataset is valuable for understanding the mammalian spillover of a pathogen with pandemic potential and migratory patterns across South America. However, there are critical issues regarding the clarity of the timeline, robustness of the conclusions, and adequacy of data sampling.

Response: Thank you for your time and effort in reviewing our manuscript and for your constructive feedback.

Major

1. The timeline of events and data collection is unclear. For example, the statement about the peak mortality of elephant seal pups between 25 September 2023 and 10 October 2023 conflicts with an earlier sentence indicating that the survey was not conducted until 10 October 2023.

Response: Thank you for bringing this up, we agree with the comment. A detailed timeline of events was included in an earlier draft but was cut to bring down the word count for submission, and this caused the narrative to be unclear. We first discovered that there was a large-scale mortality event during a visit to Punta Delgada on 8-Oct-2023, and then organized the permits, field equipment and team to conduct a proper survey and sample collection on 10-Oct-2023. During this visit and in follow up conversations, the navy personnel that operates the Punta Delgada lighthouse told us that they had first noticed unusual (although relatively minor) mortality of pups during a beach walk on 25-Sep-2023, yet they had not reported it at that time. This is why we state that the peak mortality must have occurred at some point between 25-Sep-2023 and 10-Oct-2023. The following sentence was added to clarify this (lines 66–68):

“The earliest observation of elephant seal mortality in Península Valdés was on 25-Sep-2023, when navy personnel at the Punta Delgada lighthouse noticed an unusually high number of dead pups on the beach.”

We also added a note to the legend of Table 1 to clarify that the first observation of unusual mortality was made on 25-Sep-2023, yet no counts are available, and that three site visits were made during the outbreak (10-Oct-2023, 03-Nov-2023 and 13-Nov-2023).

2. The study's modest number of viral genomes and limited sampling from wild birds raise questions about the robustness of their conclusions, particularly regarding primary transmission routes. The hypothesis that transmission occurred primarily through mammalian hosts forming a 'mammalian clade' across vast distances requires more support. A brief overview of the standard movement patterns of elephant seals and their interactions with other potential hosts would help. The high mortality rate in mammals suggests a potential alternate host (apparently healthy aquatic birds?).

While the results are presented with certainty, the discussion raises doubts about the hypothesis.

Response: We acknowledge that the lack of sampling of healthy birds during the elephant seal outbreak may be seen as a limitation to the interpretation of the findings in our study. However, we interpret findings based on the biology and ecology of the species coupled with our molecular and phylogenetic findings.

Biologically, the only species seen scavenging carcasses were kelp gulls, which we note in the manuscript and include in the discussion in reference to their potential role in spreading the virus. However, even if some infection of elephant seals occurred via infected gulls or contaminated strata (sand, rocks), it is essentially impossible for several thousand elephant seal infections to occur at the same time along 200 km of coastline from individual spill-over events. Moreover, interactions of elephant seals with gulls are mostly restricted to gulls scavenging placentas during birth. Yet pups of all ages, including weaners, were found dead. Environmental transmission through infected sand or other strata would likely also be limited by regular flooding during high tides, and salinity of seawater plus exposure to sun. The authors of this paper have extensive experience in monitoring the behaviour and natural history of the elephant seal colony in Peninsula Valdés and support the interpretation that it is biologically unfeasible for an outbreak of this magnitude and simultaneity to occur through contact with birds.

From the molecular standpoint, we found that the viruses in the marine mammal clade include the same pair of D701N and Q591K PB2 mammalian adaptations (see Figure 3). It is highly unlikely that this specific pair of mutations would independently arise over and over again in each infected marine mammal. In addition to these PB2 mutations, other mutations in PA, NS, and other genes are repeatedly found in the marine mammal viruses that would also be unlikely to arise simply by chance repeatedly each time a virus transmits from a bird to a mammal. It is far more parsimonious to infer that these groups of mutations were transmitted mammal-to-mammal. Despite the methodological challenges of making inferences about mammal-to-mammal transmission in a setting with large gaps in data, mammal-to-mammal transmission appears to be the only reasonable explanation for the observed phylogenetic clustering and mutation patterns. Conversely, for example, these mutations were not repeatedly seen in the New England harbor seal H5N1 outbreak in June 2022, where it is more likely to have been caused by multiple independent avian-to-marine mammal spill-overs. Notwithstanding, fortunately since the time of our original submission, additional H5N1 data has become available from wild birds and marine mammals in Brazil, Uruguay, Falkland/Malvinas Islands and South Georgia. When we included these new sequences in our revised phylogenetic analysis, combined with two additional full genome sequences from two dead elephant seal pups from Punta Delgada in October 2023, we found even greater support for a distinct marine mammal clade (see our updated Figure 2). First, the new H5N1 sequences from pinnipeds in Uruguay and Argentina cluster with other pinniped sequences in the marine mammal clade and also have the same mutations as the elephant seals, including D701N and Q591K, among others. Second, wild bird viruses from Brazil, Uruguay, and South Georgia are positioned outside the marine mammal clade and further supports the possibility of independent chains of H5N1 transmission in avian versus marine mammal hosts.

Overall, the phylogenetic evidence for mammal-to-mammal transmission fits with what is known about the ecology and behavior of the animals involved in this event and the spatial-temporal patterns of the outbreak. Thus, we conclude that it is unlikely that bird-to-mammal transmission alone could explain the extent, simultaneity and speed of the outbreak in elephant seals at Península

Valdés, and more broadly in pinnipeds along South America's coastlines. Here we provide some of the ecological details that were cut from the original manuscript for length:

Elephant seals spend 90% of their time at sea. Outside of the breeding season this species has a circumpolar distribution in the Southern Ocean, foraging and sleeping at sea (see <https://doi.org/10.1002/ecs2.1213>). They only come ashore for a few weeks each year to breed and a couple months later to moult (again, only for a few weeks; see <https://doi.org/10.1017/S0954102004002020>). In this context, up until September these seals would have spent that year foraging in the South Atlantic and Southern Ocean waters, only approaching land by September-October when they gathered to give birth at Peninsula Valdés (note that this is the northernmost colony for the species, all other breeding sites are located further south in Subantarctic islands). As such, opportunities for exposure to HPAI H5N1 would have been extremely limited prior to their arrival on land. In our manuscript, we briefly discuss some of the interactions with other species that might have led to exposure to HPAI H5N1 on land, namely interactions with sea lions and with scavengers such as gulls and sheathbills. During the breeding season, elephant seals gather in harems, guarded by large alpha males, that tend to keep at bay intruding sea lions and interloper elephant seal males. Although we believe that the disruption of the social structure in 2023 facilitated and increased the frequency of close and/or aggressive interactions between sea lions and elephant seals, such interactions do occur in normal circumstances. Several studies have documented and discussed how sea lions (usually young males) may enter colonies of other pinniped species and engage in aggressive interactions (<https://doi.org/10.1111/j.1748-7692.1996.tb00601.x>; <https://doi.org/10.1163/156853998792913456>; https://doi.org/10.1007/978-3-030-59184-7_9). Similar dynamics have also been reported the other way around, with young male northern elephant seals attacking other pinnipeds (<https://www.jstor.org/stable/3536542>). In our experience, these behaviours tend to be common early in the breeding season of elephant seals, when males engage in intensive aggressive behaviour while establishing territories before females arrive to give birth. In 2023, moreover, local scientists reported unusual sexual behaviour in male sea lions, including haul outs at unusual sites and interactions with female sea lions that had aborted and most likely died from HPAI infection (<https://doi.org/10.2139/ssrn.4675610>). In our visits to the field during the outbreak we recorded several dead sea lions amidst elephant seal carcasses (see Figure 1A and Table 2) and we also recorded aggressive interactions between female elephant seals with pups and intruding sea lions (see Supplementary Figure 1C and 1D).

Opportunistic scavenger birds, like seagulls and sheathbills, may interact with elephant seals post-parturition, when they approach the animals to consume placentas. However, overall, opportunities for bird-to-mammal transmission are scarce, and it would be extraordinary to claim that they suffice to explain the rapid onset and spread of the virus among elephant seals, which died simultaneously along the entire coastline of Península Valdés (about 200 km) in 2023. Data by the authors from two additional monitoring sites in the peninsula (not included in the manuscript because samples were not collected for virus screening), shows pup mortality at 97.4% and 95.7% during the same period as Punta Delgada deaths. We do not dismiss the possibility that some avian hosts in this coastal community (e.g. gulls) survived the infection, perhaps with milder clinical signs or as asymptomatic carriers, and could have played a role as reservoirs of infection at the ecosystem level. This would indeed merit future investigation. However, even if this was the case, the scarcity of opportunity for

these potential reservoirs to transmit the infection directly to each elephant seal pup or its mother, leading to thousands of nearly concurrent individual spillover events, could not reasonably translate into an outbreak of the magnitude, extent, and speed that we documented here. It is far more biologically feasible to assume that once the virus entered the elephant seal population (whether through an avian or pinniped vector) it was able to spread effectively among seals without an avian intermediary.

Due to the constraints in word limits, we cannot expand the discussion to go into extensive detail on this matter. However, we edited the discussion to try to better convey the information about the ecology of elephant seals and why we consider our findings corroborate that mammal-mammal transmission played a role in the spread of the virus. This section now reads as follows (lines 277–298):

“We posit that the high mortality rate in elephant seal pups is also consistent with mammal-to-mammal transmission, as pups are toothless and nurtured exclusively through nursing from their mothers. Contact with wild birds is minimal and could not explain the death of ~95% of all pups born in a matter of weeks, over 200 km of coastline along Península Valdés. Some newborns may have been infected before birth, as transplacental transmission of H5N1 HPAI viruses has been reported in humans⁵⁵ and high virus loads were detected in aborted sea lion fetuses^{11,30}. It could also be that mothers were infected and shed virus through their milk, infecting their pups^{56,57}. Yet, how adult female elephant seals would have been infected in the first place without mammal-to-mammal transmission presents a thornier question. Prior to arriving to give birth at Peninsula Valdés in September, the elephant seals would have spent the winter at sea in the South Atlantic and Southern Oceans⁵⁸. Feeding is an unlikely route of exposure to H5N1 HPAI viruses, since elephant seals do not eat birds or mammals, feeding instead on squid, fish and crustaceans captured in deep waters^{51,59}, and adult elephant seals will fast while on land^{51,60}. Moreover, elephant seals are pelagic and only come to shore for a few weeks to breed and later to molt, thus limiting the time window for interspecific interactions and transmission on land^{51,61}. The main interactions between birds and elephant seals involve opportunistic scavenging of elephant seals’ placental remains, molted skin and carcasses by gulls⁶⁰ (Supplementary Figure 1E), which provides more opportunities for mammal-to-bird transmission than vice-versa. In this context, it seems unlikely that bird-to-mammal transmission alone could explain the simultaneity, extent and speed of the outbreak in elephant seals at Península Valdés. Although there are still many unknowns about the precise viral transmission routes (e.g., contact, environmental, aerosol), seal-to-seal transmission seems the most plausible hypothesis to explain viral dissemination during this outbreak.”

Minor

Introduction: Line 37 needs rephrasing to reflect the likelihood of underreported widespread outbreaks in other regions: “In Europe and southern Africa, impacts on wildlife were particularly severe in seabird colonies, with losses in the tens of thousands.” For further context, please see Xie et al., Nature 2023.

Response: We thank the reviewer for this suggestion and fully agree that it is a fair point, underreporting is indeed a problem in many parts of the world and particularly in low and middle income countries. We have added the following to convey this (lines 37–38):

“Many outbreaks have likely gone underreported in other regions where influenza surveillance in animals is limited⁶.”

Results

* The first section states that signs of unusual mortality were observed around 25-Sep-2023, but the survey was not conducted until 10-Oct-2023. This gap should be addressed to clarify the timeline of events. It mentions mortality peaks and subsequent observations without specifying the methods or frequency of these observations between 25-Sep-2023 and 13-Nov-2023

Response: Our team conducted three surveys: 10-Oct-2023, 3-Nov-2023 and 13-Nov-2023 (results for all surveys are presented in Tables 1 and 2). The earlier date of 25-Sep-2023 is the reference point for outbreak onset, but this date was inferred following conversations with navy personnel at the Punta Delgada lighthouse, and there were no counts of animals at that time. Our first survey was 10-Oct-2023. We edited the text to clarify the chronology of events (lines 66–68), and added a note to the legend of Table 1 to clarify that the first observation of unusual mortality was made on 25-Sep-2023.

* The comparison of mortality rates between 2022 and 2023 lacks detail on how these rates were calculated and whether other factors could have influenced the observed increase. The section notes a 95% mortality rate by 13-Nov-2023 but does not provide information on how these figures were determined, such as the initial population size and the methodology used for monitoring mortality over time.

Response: This estimate is based on the species' behaviour and natural history, and existing 30 years of demographic information for the Peninsula Valdes elephant seal colony (see <https://doi.org/10.1111/j.1748-7692.1993.tb00424.x>, <https://doi.org/10.2307/1383185> and <https://doi.org/10.1111/mms.13101>). As summarized in the section “Study species” in methods, elephant seal pups remain on the beach where they were born while nursing from their fasting mothers, for approx. 22 days. During the nursing period, the pups always remain very close to their mothers, and vocalize vigorously if separated by more than a few meters. Weaning occurs abruptly when the mother departs to sea, after which the pup (now weaner) remains on the beach, fasting for >5 weeks to continue its physiological development (e.g. acquiring diving capacity, <https://doi.org/10.1111/j.1748-7692.2001.tb01302.x>). During this period (3 weeks nursing and 5+ weeks as weaners), their ability to galumph or swim is extremely limited. Furthermore, the Punta Delgada area and most Peninsula Valdes beaches are enclosed by steep cliffs, which prevents pups from galumphing or crawling to adjacent beaches. In this context, if newborn pups seen on the beach in early October are no longer present on the same beach approximately one month later, it is safe to presume that they died.

October 1st to 5th is the peak of the breeding season at Peninsula Valdes, when the maximum number of females and their pups are present, plus weaners accounting for females already departed. Thus, annual censuses are conducted at this time, weather permitting. In 2023, our first dedicated survey was on 10-Oct-2023, and we counted 235 living pups/weaners and 570 dead pups. One month later, on 13-Nov-2023, we counted 38 living pups/weaners. From this, we estimate that 805 pups were born at Punta Delgada in 2023 (235+570=805), but only 5% survived (38/805=0.0472), hence the 95% mortality estimate. Moreover, a similar, yet slightly higher, mortality rate (96%) is obtained when considering the expected survival of pups/weaners (n=1005) based on

data from 2022. It should be noted that during field visits to Punta Delgada in October and November 2023 we saw and photographed hundreds of pup carcasses on the beach. However many were covered by sand dunes, and we also noticed that some areas that had carcasses in previous visits were now clear, as the carcasses had been moved by tides. There was also extensive scavenging and carcasses were quickly decomposing and falling apart. For this reason, we felt that counting carcasses on the beach in November was not a reliable strategy to estimate the mortality rate, which is why we preferred the approach described above.

The following text was added to clarify how the mortality estimate was made (lines 431–434):

“Because pups and weaners have extremely limited mobility and cannot leave the beach for >7 weeks until they have finished their development, pup survival in 2023 was estimated by dividing the number of living pups/weaners counted on 13-Nov-2023 by the total number of pups/weaners counted on 10-Oct-2023.”

* Figure 5 is well-constructed and informative. However, the accompanying text should emphasize the limitations of the study, including intermittent surveillance and modest sample size.

Response: Thank you for this suggestion. The following text was added to the figure caption:

“Note that there are significant differences in surveillance strategies among countries that may produce gaps or distortions in the geographic distribution of H5Nx HPAI detections and the presumed pathways of virus spread.”

REVIEWER #2

In this paper, Uhart and coauthors describe a massive outbreak of avian influenza in elephant seals and put forward strong evidence for mammal-to-mammal transmission of influenza viruses. The authors observed many dead southern elephant seal pups at Península Valdés compared to the previous year. The authors also observed a breakdown in the social and breeding structure with a lack of adult seals. Pups presented with symptoms that suggested avian influenza. The lack of feeding when out of the water and the limited possible interactions between pups and birds strongly suggested mammal-to-mammal transmission. The authors also noted that many terns also died and showed symptoms of influenza. The peak of tern mortality occurred several weeks after the peak of seal mortality. Sequencing was performed from 1 pup, 1 subadult male that hauled out at an aberrant location after the outbreak at Península Valdés and 3 terns. The authors show that their sequences and other mammalian sequences from Argentina/Chile/Brazil form a monophyletic clade which is separate from the wild bird sequences from inland Argentina. They show that there were 3 jumps from birds into mammals though only one of these jumps led to sustained spread with multiple recorded sequences. The marine mammal clade contains known mutations associated with mammalian adaptation which have been maintained in the terns. The authors suggest that influenza spread via mammal-to-mammal transmission down the west coast of South America to the tip of Chile before spreading up the East Coast of South America through Argentina, Uruguay and Brazil. Finally, the authors claim that there was a lower evolutionary rate in marine mammals compared to the avian virus with all segments having strong purifying selection.

The paper is clearly written and strongly suggests mammal-to-mammal transmission. Figure 5 is an

especially well made and useful figure. However, the paper would have benefited from more data to confirm mammal-to-mammal transmission.

Response: Thank you for your time and effort in reviewing our manuscript and for your constructive feedback.

The paper presents lots of circumstantial evidence of a substantial avian-flu outbreak in the seals but unfortunately there was only one actual new sequence gathered from a seal pup. Despite the suggestion of many adult deaths, no sequences were gathered from adult seals and the authors suggested that most deaths would happen at sea.

Response: Please see the response to Reviewer 1, which describes our successful effort to obtain whole-genome sequences from two additional elephant seal pups since submitting our original manuscript, which increases the size of our dataset. With regards to the mortality of adults, we were very keen to sample more adult seals, however all the adults we found dead at Punta Delgada were too decayed to be sampled. In addition, the local animal health authority tested one adult elephant seal at Punta Tombo, Argentina (non-breeding area approximately 190 km southwest from Punta Delgada), which was positive for H5N1 (WAHIS event 5189 outbreak OB_124986; unfortunately sequencing was not performed). We have now added this additional adult seal report to the manuscript.

While overall data is limited, the confirmed case in an adult seal at Punta Tombo combined with our data from Puerto Madryn/Golfo Nuevo, corroborate that some level of adult mortality due to H5N1 HPAI occurred outside of the breeding colonies. The magnitude of the adult mortality, however, is difficult to estimate. We counted at least 35 adult carcasses at Punta Delgada, which is definitely far above the baseline (for instance, in 2013, 2015 and 2022 no adult mortality was seen at this site, see Table 1). However, we suspect that the actual death toll must have been considerably higher. Notwithstanding, the data currently available is not sufficient to provide a reliable estimate, and experts concur that the best adult mortality/survival estimate will have to be produced in the upcoming seasons, by counting returning female elephant seals and by evaluating population trends. Fortunately, demographic data for the Peninsula Valdés colony exists for the last 30 years, based on annual censuses, and there are plans for colony monitoring in the coming years (see <https://doi.org/10.1111/j.1748-7692.2012.00585.x>).

We added the following text to mention the detection of HPAI H5N1 in an additional adult elephant seal, which helps contextualize our suspicion of at-sea mortality of adults (lines 260–262):

“Furthermore, a dead adult male elephant seal at Punta Tombo (non-breeding area approximately 190 km southwest from Punta Delgada) was confirmed positive for H5N1 HPAI virus by national authorities in mid-September 2023⁵⁰, prior to the onset of the Península Valdés outbreak.”

There are many unanswered questions about this outbreak. How did the first seal get infected? The authors suggest that it was unlikely through diet as seals eat squid, fish and crustaceans and therefore were not consuming animals infected with influenza. If the seals were infected by sea lions, how was the first seal infected if sea lions interacted more only following the breakdown of social structure? Also, why were the seals infected in October when the virus past through the area in sea lions several months before? Were the infections only seen when the seals hauled out?

Response: The fact that the mass mortality was only discovered when hundreds of pups had already died precludes us from making a decisive statement about how the outbreak began. We can only infer based on the epidemiological circumstances, the natural history of the species involved, and the insights provided by our genomic data.

Although we believe that the disruption of the social structure facilitated and increased the frequency of close and/or aggressive interactions between sea lions and elephant seals, such interactions do occur in normal circumstances. Several studies have documented and discussed how sea lions (usually young males) may enter colonies of other pinniped species and engage in aggressive interactions (<https://doi.org/10.1111/j.1748-7692.1996.tb00601.x>; <https://doi.org/10.1163/156853998792913456>; https://doi.org/10.1007/978-3-030-59184-7_9). Similar dynamics have also been reported the other way around, with young male northern elephant seals attacking other pinnipeds (<https://www.jstor.org/stable/3536542>). In our experience, these behaviours tend to be common early in the breeding season of elephant seals, when males engage in intensive aggressive behaviour while establishing territories before females arrive to give birth. Also, a study by local scientists reported abortions in females and unusual sexual behaviour in male sea lions, including haul outs at unusual sites and interactions with female sea lions that had aborted and most likely died from HPAI infection (<https://doi.org/10.2139/ssrn.4675610>). Finally, in our visits to the field during the outbreak we recorded several dead sea lions amidst elephant seal carcasses (see Figure 1 and Table 2) and we also recorded aggressive interactions between female elephant seals with pups and intruding sea lions (see Supplemental Figure 1C and 1D).

With regards to why seals were only infected in late September while sea lions had been affected nearby since August, this is related to the timing in which the elephant seals haul-out. Elephant seals spend 90% of their time in the ocean, and only haul out a couple times a year, for a few weeks each time, to breed and to moult. During the breeding season at Peninsula Valdes, although the first elephant seals start to haul-out in the third week of August, their numbers only increase substantially by mid-September, reaching their peak on the last week of September and the first week of October. Since the births also occur around that time (females give birth 5-6 days after hauling-out), the first week of October is the peak of beach occupation by elephant seals (this is why the annual population surveys are done that week). The figure below illustrates this timeline of events (reproduced from <https://doi.org/10.1111/j.1748-7692.1993.tb00424.x>):

We understand that elephant seals have a unique and extreme life cycle, and journal readers may have the same questions if they are not familiar with the natural history of southern elephant seals. The Methods subsection “Study species” is intended to summarize the key information that readers would need in order to understand/interpret our findings, however perhaps we have not provided sufficient information on the phenology of the species. Considering the reviewer’s comments, we have added the additional text to this section, which now reads as follows (lines 402–420):

“Southern elephant seals are widely distributed in Subantarctic islands, with a single continental colony at Península Valdés, Patagonia, Argentina (representing ~5% of the global population)⁷⁷. The species has a well-defined annual life cycle, which we summarize as follows based on studies at Península Valdés^{51,61}. Adult (and subadult) males and females haul-out in late August and early September, with alpha males establishing and defending harems (median 11–13 females per harem, with a maximum of 134 females); subordinate males are chased away but remain along the margins of harems. The number of adults on the beach rapidly increases during the second half of September, reaching its peak by the end of that month. Most females are pregnant when they come ashore, giving birth within 5.7 ± 1.9 days after arrival (80% of pups are born by 2 October). Pups are toothless and will nurse for 22.4 ± 1.7 days; during this period the females will fast and remain with their pups, under the protection of the alpha male. Copulations will begin 20.3 ± 2.1 days after parturition, i.e. shortly before females wean their pups. The female then abandons the pup and returns to the sea to forage; on average, females spend a total of 28.2 ± 2.5 days ashore, fasting. Males also fast on land and will abandon the beach approximately at the same time as females; adult seals are nearly absent by mid-November. The number of weaned pups will increase during the second half of October, reaching its peak by the end of November. Weaners will remain on the beach for >5 weeks, fasting while they complete their physiological development and are ready to go to sea to forage. Juveniles and adults will return to the beaches later in the season to undergo molt, with juveniles molting earlier (November to January) than subadults and adults (December to February).”

The roles of sea birds in transmission is also extremely unclear. How did terns get infected? Were other seabirds involved in transmission to or from the seals? The lack of contemporaneous

sequencing of birds hinders the ability to really understand the dynamics of influenza in this ecosystem. Is there a possibility that the first seals were infected from environmental contamination from sea birds?

Response: The first seals are unlikely to have been infected by environmental contamination from seabirds because the viruses from marine mammals in Argentina are so genetically similar to those from marine mammals in Chile and Peru, with exactly the same constellation of mutations (see Figure 3), as we described in our response to reviewer 1 above. While there are few data from wild birds in Argentina apart from the four H5N1-infected terns we identified and sequenced, our study greatly benefits from the larger context provided by the wild bird data available from Peru, Chile, Brazil, Uruguay and South Georgia, which allowed us to identify clear signature mutation patterns, including known mammalian adaptations, that are consistently different between marine mammals and wild birds across South America. It would be highly unlikely that the exact same mammalian adaptations, along with other mutations, would occur repeatedly in marine mammals if the seals were infected by environmental contamination with a wild bird virus.

Broadly speaking, the chronology and geography of sea lion deaths and HPAI H5N1 detections in Argentina seem to point to sea lions as the main vehicle for the spread of marine mammal H5N1 viruses. Moreover, sea lion deaths mark the southward path of the virus along the Pacific coast of Chile. The southernmost continental cases were two sea lions in c on the Beagle Channel, across Tierra del Fuego Island, in mid-June. Then the first cases in Argentina were detected on the Atlantic coast of Tierra del Fuego, near Rio Grande, the first week of August. And from there cases began to be reported northward, along the coast of Argentina and then Uruguay, also in sea lions, throughout August. This path is marked with the blue arrow in Figure 4. Unfortunately, Chile confirmed HPAI H5N1 infections but could not produce genome sequences from sea lions at the most austral locations. Notwithstanding, the genomic evidence in our study (phylogeny, phylodynamics, mammal-adaptation mutations, genome-wide mutation rates) also point to the “marine mammal clade” strains being transmitted and evolving primarily in mammalian hosts. This said, we don’t know whether seabirds may have played an additional, “hidden” role in the epidemiology of the elephant seal outbreak.

In a previous paper (<https://doi.org/10.3201%2Fcid3004.231725>), we reported on the HPAI H5N1 outbreak at the Punta Bermeja sea lion rookery, approximately 180 km north from Punta Delgada, in late August 2023. In this outbreak, we found a small number of birds (a tern and a grebe) that had died and were H5N1 positive, but the bird mortality was minimal compared to the 800 sea lions that died at this site (roughly 25% of the rookery). It is plausible that birds infected at Punta Bermeja (or at other sea lion rookeries) could have contaminated the environment at Península Valdés and caused the first elephant seals to become infected. However, if we consider that mammal-to-mammal transmission is possible among elephant seals (as our epidemiological data and the natural history of this species suggest), it would also make sense that sea lions could have simply transmitted the infection to the elephant seals without the need for an avian/environmental intermediary.

The question of how the terns became infected is also something we would like to address, but do not yet know. Birds that prey or scavenge on mammals (such as gulls, sheathbills, skuas, giant petrels, fulmars, vultures, raptors, etc.) have certainly had plenty of opportunities to be exposed, but terns do not engage in such behaviour. Our best guess is that there is perhaps an intermediate/bridge host such as gulls that links the pinnipeds to the terns and other coastal birds (such as the grebes at Punta Bermeja). Gulls have been observed scavenging on sea lions and

elephant seals throughout the coastal Argentine outbreaks, and they also frequently interact with terns (e.g. predation on eggs and chicks, nesting in close proximity, etc.). Yet, the tern deaths at Punta Delgada occurred prior to the onset of the tern breeding season, and there are no tern breeding colonies at Punta Delgada. Thus, we would not have expected for interactions with gulls to have been common at that point.

Sampling living seabirds at the time of the outbreaks was not feasible due to concerns with the risk of introducing the virus to new areas or facilitating transmission between species. Moreover, in places such as Punta Delgada there were hardly any living animals left, and it would've been ethically questionable and counter-productive from a conservation standpoint to add further stress and disturbance to the few remaining survivors that still had a chance to produce offspring. Should the situation be less dramatic in the upcoming 2024 season, we will aim to sample living seabirds in Patagonia, including mildly symptomatic and asymptomatic animals – and hopefully this will help us to address some of the remaining questions.

We edited the text to mention the possibility of spread by seabirds and provide more context on this matter (lines 299–322):

*“How the virus was first introduced to the elephant seals when they arrived at the beaches is unclear, but sea lions appear to be the most likely source. Notably, the epidemic path of HPAI along coastal Argentina left virtually no rookery or stretch of beach without dead or symptomatic sea lions from south to north^{30,62}, and then progressed to neighboring Uruguay and Brazil^{31,32}. This unrelenting spread along the Atlantic coast mirrored that seen along the Pacific coast, with the common denominator being infected sea lions^{11,27} (Figure 4). South American sea lions regularly visit multiple rookeries and haul-outs, sometimes interacting aggressively with other pinnipeds^{63,64}. At Punta Delgada, we observed numerous sea lion carcasses amongst dead elephant seals (Figure 1A, Table 2) and witnessed aggressive interactions between sea lions and elephant seals (Supplementary Figures 1C and 1D). Government veterinarians who monitored sea lion rookeries in Argentina noted that some animals showing clinical signs of HPAI survived for several days and often abandoned the rookeries while ill (Veronica Sierra, pers. comm.). It is plausible that these sea lions visited different sites during their convalescent period, including elephant seal colonies, and may have played a key role in the spread of H5N1 viruses. Considering the occurrence of a multi-species outbreak with sea lions, fur seals and seabirds one month prior in the region³⁰, it is conceivable that birds infected in that outbreak also contributed to the spread of the virus to the beaches occupied by elephant seals in Península Valdés. In addition, mammal-to-bird spillovers do not seem improbable given the frequent observations of gulls and other avian scavengers feeding on sea lion and elephant seal carcasses in Argentina (Supplementary Figure 1E). It is interesting to note that the earliest H5N1 HPAI detection in the Falkland/Malvinas Islands was that of a marine mammal clade virus in a southern fulmar (*Fulmarus glacialis*) in late October 2023⁶⁵. Fulmars are known to occasionally scavenge on birds and mammals^{66–68}, and this detection raises the possibility that scavenging procellariiform birds could also play a role in the spread of these viruses. On the other hand, it is still unclear how terns (which are not scavengers) became infected, and further studies may help to clarify whether other birds (e.g. gulls) played a role as bridging hosts between pinnipeds and terns.”*

I think that this paper has the potential to be extremely important in demonstrating that there is sustained mammalian spread of H5. The natural history information of the seals very strongly suggests this and I am convinced there is mammal-mammal spread. However, it would be extremely helpful if the authors manage to acquire some more data to bolster their conclusions perhaps by combining data with the group working in South Georgia.

Response: Thank you for highlighting the relevance of this subject. We agree that it would be helpful to collaborate with other groups, and we have actively engaged with researchers in the international community, including OFFLU, since the onset of the outbreaks in Patagonia. Since we originally submitted our manuscript, our colleagues from South Georgia have recently uploaded new sequences to public databases and updated the preprint presenting their more recent findings (<https://www.biorxiv.org/content/10.1101/2023.11.23.568045v2>). In brief, their results reveal that the HPAI H5N1 strains detected in South Georgia (including in elephant seals and fur seals) do not belong to the “marine mammal clade” that we discuss in our paper. Instead, the outbreaks in South Georgia were closer to the HPAI H5N1 strains that circulated months earlier in poultry and wild birds in Argentina and Uruguay. We re-ran our phylogenetic analyses and edited the manuscript to take this new information into account. Specifically, the part of the Discussion that mentions South Georgia now reads (lines 362–367):

“Significant mortalities of southern elephant seals and Antarctic fur seals (Arctocephalus gazella) have been attributed to H5N1 HPAI in South Georgia^{65,78}, however the viruses involved do not belong to the marine mammal clade identified in this study, clustering instead with avian viruses from inland Argentina⁶⁵. Considering that 95% of Antarctic fur seals⁷⁹ and 50% of southern elephant seals⁸⁰ breed in South Georgia, these populations could be at great risk if the marine mammal clade viruses spread there in the future.”

Major Concerns

1. I am concerned by the dN/dS data presented in Supplemental figure 7. The legend does not clearly state what the box and whisker plots shows or what n is. It seems odd to me that all the segments have such identical values. You might expect that there might be positive selection on some segments or that the polymerase segments are less tolerant of mutations. The data is referred to as <0.3 on L178 when it is clearly <0.2.

Response: There were problems in the dN/dS analysis, and we opted to remove Supplemental Figure 7 and perform this analysis in the future when we have more data from marine mammals to compare against avian hosts.

2. Figure 3 only analyses South American avian sequences but there is no reason not to include global sequences which might be more intensively sampled. Is it possible that reassortment has affected the results of this or other analyses in any way? This should be commented on. The global results for each segment could be included here as a comparison to the South American concatenated tree which would ensure that reassortment had not affected the results.

Response: The reviewer raises an excellent point, as reassortment occurs frequently in birds and can result in differing phylogenetic constructions for different IAV segments. Fortunately, prior published studies from South America have already performed this global analysis for all eight genome segments for closely related H5N1 viruses from Peru (see Figure 2 in Leguia et al., <https://doi.org/10.1038/s41467-023-41182-0>). Since our viruses are so closely related to the Peru

H5N1 viruses published by Leguia et al., and belong to the same B3.2 introduction from North America (Supplementary Figure 3), we opted not to repeat their analysis and instead cite their findings. However, the reviewer raises an important point that we have not observed any additional reassortment events between the B3.2 viruses introduced from North America and the LPAI South American lineage that is enzootic in Argentina, Chile and Peru. The text was edited to convey this as follows (lines 340–342):

“To date, no reassortment has been observed between H5N1 HPAI viruses in South America and LPAI viruses belonging to the South America lineage that circulate enzootically in aquatic birds in Argentina, Chile and Peru^{11,73}.”

3. I would ideally like more data to confirm that there is mammal-mammal transmission but if not the discussion should make clear the limitations of the data and what data needs to be collected to confirm the hypothesis.

Response: Other data such as transmission experiments would indeed have been useful to demonstrate mammal-mammal transmission, however our research permits and concerns for biosecurity precluded us from collecting live virus. Based on the data and samples that we were authorized to collect, we have attempted to explore all possible perspectives that might help clarify how the virus spread and whether mammal-mammal transmission did occur, however there are inherent limitations to what we could do without live virus. Nevertheless, we would point out that transmission experiments using a virus strain from a human case in Chile that is closely related to the strain we found in Argentina, and which is part of the same phylogenetic tree branch and shares many mammal-adaptation mutations, demonstrated that the virus was able to be transmitted directly among ferrets (<https://doi.org/10.1080/22221751.2024.2332667>). In addition, after the submission of the manuscript we obtained whole genome sequences of viruses from oronasal, tracheal, lung, brain and rectal swabs of one elephant seal pup. This revealed that the same cassette of mammal-adaptation mutations was present in all examined samples, corroborating that there is onward transmission of these mutations (i.e. they are not *de novo* mutation events).

Nevertheless, we agree that there are many open questions and that, while we feel our evidence is very compelling, we do not provide *in vivo* proof of mammal-mammal transmission. To ensure that readers are aware of the limitations of the study and to indicate what data would need to be collected to further corroborate the hypothesis, the text was edited to read as follows (lines 349–357):

“Despite gaps in the available data, our epidemiological and phylogenetic results support the hypothesis that the spread of viruses from the novel marine mammal clade in South America has occurred via mammal-to-mammal transmission, but as with any hypothesis, this is subject to revision as more data become available. Additional studies on the virus prevalence, shedding, and genome in different potential hosts within the coastal wildlife of Patagonia, especially for species that were already shown to be susceptible to H5N1 HPAI viruses in other regions (eg. skuas, gulls, petrels) would be helpful to identify if there are asymptomatic reservoirs of infection and bridge hosts. Furthermore, the recovery of live virus for pathogenesis and transmission studies would also be valuable to demonstrate how these strains behave in mammalian experimental models.”

Minor Concerns

4. In the introduction, please add some focus on mammal-mammal transmission.

Response: Due to word limits we cannot expand the introduction much further, and we feel that it is essential to maintain the chronological background of how H5N1 HPAI viruses arrived to South America and affected marine mammals. However, we added the following sentence to provide additional background on mammal-mammal transmission (lines 43–48):

“Until recently, it was generally considered that H5N1 HPAI infections in mammals were largely limited to terrestrial carnivores that consumed or otherwise interacted with infected birds^{16–18} and these viruses generally showed limited airborne transmissibility in mammalian models^{19–21}. During the 2021–2022 panzootic, H5N1 HPAI caused episodic mortality of pinnipeds and cetaceans in Europe^{22,23} and North America^{24–26}, but it was only upon reaching the Pacific coast of South America that the virus demonstrated an ability to cause large-scale mortality in marine mammals^{11,27}.”

5. L92 a graph showing mortality of terns would be nice.

Response: Our data on tern mortality is already presented in Table 2, and we feel that our temporal resolution is not sufficient to make a compelling graph.

6. L110 exactly how the samples were pooled is not clear here and elsewhere. Please make sure it explicitly states what was pooled and what was sequenced.

Response: This information is provided in Table 3 (previously Supplementary Table 3) and Supplementary Table 2, and the text was amended to direct readers to those tables.

7. The phylogeny in figure 2 should include how the clades are related to each other. They should not be presented as separate clades.

Response: We thank the reviewer for this suggestion and note that there was an error in the submission of Figure 2, which cropped the left margin of the figure and as a result did not include the branch showing how the clades are related. We re-ran the phylogeny including additional H5N1 viruses from Argentina, Uruguay, Brazil, Falkland/Malvinas Islands and South Georgia, and the new version of Figure 2 now adequately shows how the avian and marine mammal clades are related to each other.

8. Figure 3a with only a single human and sanderling case there is no need to present a value. Other values should be presented with a standard error.

Response: Nodes often have multiple location states, with probabilities adding to 1. It is therefore common for a singleton virus (e.g., A/Chile/25945/2023) to have a set of posterior probabilities from different locations (e.g., 0.3 location A, 0.5 location B, 0.2 location C). The only reason the value is 1.0 for the human and sanderling is because in this case there is high support for the inferred location states in the branches leading up to these samples. However, a reader cannot assume this low level of uncertainty in all contexts, so it is informative to include the actual value in the figure. There isn't enough space in Figure 2B to include the uncertainty, so 95% HPD (highest posterior density) values

are presented in the new Supplementary Figure 7.

9. L170 fix 12.

Response: Corrected.

10. L183 add 'compared to'.

Response: We added "compared with", based on grammatical correctness: The phrase "compared with" is used to compare similar things, while the phrase "compared to" is used to compare dissimilar things.

11. L290-291 is there any evidence for this?

Response: No direct evidence, unfortunately. All we can say at this point is what we have already mentioned in the manuscript, namely that (a) infected/dead/ill sea lions were reported throughout the entire coast of Argentina, (b) some of these symptomatic individuals were seen to abandon the rookeries, and (c) living and deceased sea lions were seen at Punta Delgada and interacting with elephant seals. We are not aware of any studies in Argentina at the moment tracking sea lions (e.g. satellite or GPS trackers) or marking them (e.g. flipper tags or fur dyes), hence we have no way to determine whether the individuals we saw at Punta Delgada had previously been at a rookery with H5N1 mortality. This said, there are several sea lion rookeries at Peninsula Valdes that were affected by H5N1 by late August. So, there were infected sea lions in relative proximity to elephant seal haul outs, including Punta Delgada, making incursions by sick sea lions feasible and likely. Since we cannot ascertain whether sea lions found at the elephant seal mortality sites were positive, we have phrased this statement as "*It is plausible that (...) may have played a role*", i.e. we are indicating a possibility but not making a claim on likelihood.

12. L304 please comment on why there might be a lower rate of evolution. Perhaps a comparison to work on cows would be appropriate here.

Response: Data made recently available reveals that the evolutionary rate for cattle appears to be avian-like with 6.23×10^{-3} substitutions/site/year (95% highest posterior density (HPD), 5.29×10^{-3} - 7.19×10^{-3}) (<https://doi.org/10.1101/2024.05.01.591751>). We now cite this paper. We believe that the lower evolution rate in marine mammals may relate to the artificial densities of farmed animals, dairy cows or chickens, do not resemble free-ranging wildlife situations, with the rare exception of colonial breeding seabirds, in which higher evolutionary rates have indeed been seen. The following text was added to convey this (lines 335–340):

"Of note, the rate of evolution of the marine mammal viruses is not only lower than that of H5N1 viruses circulating in birds in South America, but it is also considerably lower than that of H5N1 viruses circulating in cattle in the USA (6.2×10^{-3} ; 5.3 – 7.2×10^{-3} 95% HPD)⁷². This could relate to the high densities of susceptible hosts in farmed animals (dairy cows or poultry) and colonial seabirds providing opportunities for rapid and sustained transmission."

13. L311 jumped.

Response: Corrected.

14. L340-350 please tone this down as there is little risk currently shown to public health.

Response: The text was edited and now it reads as follows (lines 375–387):

“From a public health perspective, mammal-to-mammal transmission could be a stepping-stone in the evolutionary pathway for these viruses to become capable of human-to-human transmission⁸⁴. As mentioned previously, some of the mutations found in the strains of the marine mammal clade are already known to be of concern. In particular, the mutation D701N in PB2 has been shown to compensate for the lack of the E627K mutation in PB2 in terms of improved viral growth in mammalian cells and enhanced aerosol transmission of H3N2 and H5N1 viruses⁸⁵. The fact that the H5N1 HPAI virus detected in a human case in Chile belongs to the marine mammal clade and is transmissible among ferrets^{54,86} highlights the potential risk to public health. In addition, the possibility of zoonotic strains resulting from the reassortment between HPAI viruses and other IAV strains infecting pinnipeds is also of concern^{42,87}, especially since pinnipeds are known to occasionally host human-like IAV strains^{44–46}. However, the phenotypic effects of mutations in other gene segments found in the H5 viruses from our study (Supplementary Table 4) are not yet known, and further research using in vivo mammalian models is needed to determine whether they can enhance the virulence and/or transmission.”

15. L351-2 please tone this down as many compartments and species remain unaffected by flu.

Response: We have extensively edited the conclusion paragraph and it now has a different focus, reading as follows (lines 388–399):

“In conclusion, the world has seen a drastic increase in the number of H5N1 HPAI detections in mammals since 2023, including unprecedented outbreaks such as the one reported here. Amidst growing evidence that mammal-to-mammal transmission played a role in H5N1 HPAI outbreaks in dairy cows in North America^{56,88} and in fur farms in Europe^{89,90}, the outbreak among elephant seals in Península Valdés represents another case where mammal-to-mammal transmission was potentially involved in the spread of H5N1 HPAI infections, this time in free-ranging wildlife. Genetic drift and shift in IAVs are stochastically-driven phenomena⁹¹, and mutations that increase transmissibility between mammals are more likely to occur in mammals than birds^{92,93}. Therefore, the recent increase in H5N1 HPAI circulation in mammals is a warning that should not be ignored. Moving forward, HPAI management requires holistic strategies that recognize the interconnectedness of human, animal, and environmental health and safeguard biodiversity, promote sustainable practices, and enhance resilience globally to emerging infectious diseases.”

16. L428 how was this model chosen

Response: We selected a GTR model of nucleotide substitution with gamma distributed rate variation among sites because RNA viruses like influenza evolve so rapidly ($\sim 10^{-3}$ substitutions/site/year) and require a flexible model that allows for high rate variation. As a result, GTR+G is the most widely used model for IAV evolution.

17. Is it possible to add extra years of baseline to Table 1

Response: Unfortunately, the annual census did not include Punta Delgada from 2016 to 2021, and we also don't have data for 2014. We edited Table 1 to present data from the annual censuses of 2013 and 2015, which provide information that is generally very similar to that of 2022. One caveat is that because the elephant seal population of Península Valdés had been steadily increasing prior to the HPAI outbreak (1.0 to 3.4% increase in adult females per year; <https://doi.org/10.1111/j.1748-7692.2012.00585.x>), this means that the data from 2013 and 2015 show a noticeably smaller population size compared to 2022 and 2023. We edited the legend of Table 1 to convey this and avoid any confusion.

18. Supplementary Figure 3 is not helpful without labels and should be combined with another figure with labels if used.

Response: We now include tip labels and label important sections of the tree (e.g., "marine mammals," "poultry," "South Georgia." We also include the HA tree file with all tip and node labels in the GitHub repository (<https://github.com/mostmarmot/ArgentinaH5N1>).

19. Please be explicit when comparing South American mammals to birds whether terns falling in the mammalian clade and mammals falling in the avian clade have been excluded.

Response: The reviewer raises a great point. Our original manuscript specified that the HSLC analysis excluded human and avian viruses from the "marine mammal clade" and also excluded marine mammal viruses from the "avian clade". The Methods section was edited to clarify that this also applies to the terns from Argentina, i.e. that they are excluded from the "marine mammal clade" in the HSLC analysis (lines 513–517):

"For the HSLC analysis, any singleton avian and human viruses positioned in the marine mammal clade (likely representing transient dead-end spillovers) were excluded to ensure monophyly (the four viruses detected in terns in this study were also excluded). Similarly, any singleton marine mammal viruses positioned in the major avian clade (which also likely represent transient dead-end spillovers from birds to marine mammals) were excluded."

This is also made clear in the Results section (lines 178–181):

"Using a HSLC, the estimated rate of evolution in the marine mammal clade (human and avian viruses excluded) was ~2-fold lower (2.5×10^{-3} ; $2.0\text{--}3.0 \times 10^{-3}$ 95% HPD) than the avian rate (5.4×10^{-3} ; $4.9\text{--}5.9 \times 10^{-3}$ 95% HPD), which includes wild birds and poultry but excludes spillovers into mammals (Figure 2C)."

The caption of Figure 2C (previously Figure 3B) was also edited to clarify this and now reads as follows:

"Posterior distributions of evolutionary rates (substitutions per site per year) inferred for the complete virus genome (all positions) and for only the third nucleotide position for H5N1 (2.3.4.4b) in South America, partitioned into two host categories: marine mammal clade (excluding any human and avian viruses) and wild bird/poultry clade (excluding any marine mammal viruses)."

Reviewer #2:

The authors have mostly addressed the comments.

The paper itself makes two significant claims – 1. Mass mortality of seal pups due to avian influenza. 2. Mammal-mammal transmission of avian influenza. I believe both of these claims to be true but the authors must be very careful in their claims as the evidence they present is mostly circumstantial. For claim 1, the sequencing evidence confirms that 3 pups had influenza A and given the timing of the virus and symptoms, it is clearly reasonable to assume this is what caused the deaths (even if no healthy individuals were swabbed to ensure they did not have the virus and only a few pups were confirmed to have flu.) For claim 2, the evidence is more circumstantial. There is clearly a large mammal clade, and the biology of the elephant seals means that they are unlikely to have acquired the virus except from conspecifics. However, questions on transmission still remain unanswered. It is possible that a sea lion infected the first seal but there is no particular evidence to support this. We do have evidence of birds being infected with mammalian-adapted sequences and we do not have contemporaneous bird sequence data from near the seals except for the terns which show that large scale infection of birds with mammalian adapted virus is possible. It is therefore possible that the seals could have been infected by an environmental or avian source. I think this is unlikely but it is clearly possible in some cases e.g. in the Falkland islands. The continental spread of the mammalian adapted sequence could easily have been facilitated by birds feeding on corpses leading to a cycle of transmission between birds and pinnipeds.

RESPONSE:

Thank you for taking the time to review our paper. While our study does leave some open questions (as do nearly all published studies), we believe it is unreasonable to characterize our evidence as "mostly circumstantial".

With regards to claim 1, that the mass mortality of elephant seals was caused by high pathogenicity avian influenza (HPAI), we demonstrated the presence of avian influenza viruses in the brain and other tissues of multiple affected individuals through RT-qPCR and produced whole genome sequences to demonstrate that the virus has the genomic characteristics that are determinant of high pathogenicity. Moreover, we also showed that identical virus was present in multiple organs of a single elephant seal, which reflects systemic infection, characteristic of HPAI. This is direct demonstration of the proposed causative agent, not circumstantial evidence. Moreover, in all reported wildlife mortalities where observed deaths exceeded by far expected mortality rates and HPAI was identified in a subset of affected individuals (with all individuals sampled being confirmed positive), the mortality event was reported to be due to HPAI. Nobody has tested thousands of animals in a single event when a virus as deadly as HPAI H5N1 has been diagnosed with matching epidemiological data.

With regards to claim 2, that mammal-mammal transmission played a significant role in the outbreak, while we did not perform transmission experiments with the seals, we have presented both epidemiological and molecular data, including phylodynamic and mutation analyses, as evidence. This evidence provides a strong and non-redundant basis for our inference on mammal-mammal transmission, and is far from circumstantial. Of course some questions on transmission remain unanswered. Yet, we believe our manuscript conveys these questions and uncertainties in an honest manner. We have revised the text and made edits where necessary (changes tracked in the manuscript) to ensure that any specific points that are speculative hypotheses are clearly communicated as such. For instance, with regards to the hypothesis that sea lions might have been

the initial source of infection, we should clarify that while this is a reasonable hypothesis given the widespread distribution of H5N1 HPAI outbreaks in sea lions in Argentina at the time, we are not ruling out that birds could also have played a role. Unfortunately, the extremely limited data currently available from the Falkland/Malvinas Islands precludes any meaningful insight that could assist the interpretation of our data – only one H5N1 genome sequence is available for the islands, and no epidemiological data has been released on the potential circulation of the virus in pinnipeds in that archipelago.

Regrettably, it is impossible for us to dismiss all the "what if" thought experiments or "is it possible that" discussions on pathways of transmission. But we would argue our manuscript puts forth a robust argument as to why mammal-to-mammal transmission is the most reasonable explanation to the epidemiological and molecular evidence. While we understand the reviewer's hesitation to abandon the paradigm that mammal-to-mammal transmission is uncommon and is generally inefficient for HPAI viruses, we would highlight that there is growing consensus that mammal-to-mammal transmission has played a significant role in the recent spread of H5N1 HPAI viruses, including among marine mammals in South America. An example is the article published recently in Nature by Peacock and colleagues which explicitly makes the case that there is growing evidence that these viruses are opening new evolutionary pathways that involve efficient transmission among mammals (<https://www.nature.com/articles/s41586-024-08054-z>).

In L352, please be more explicit about the cause of uncertainty rather than saying but as with any hypothesis, this is subject to revision as more data become available. What are the other possibilities and what data would you like?

RESPONSE:

The alternative hypothesis would be that the outbreak was driven by individual bird-mammal transmission, and the causes of uncertainty relating to these two hypotheses are discussed extensively in the following sections of the discussion:

"We posit that the high mortality rate in elephant seal pups is also consistent with mammal-to-mammal transmission, as pups are toothless and nurtured exclusively through nursing from their mothers. Contact with wild birds is minimal and could not explain the death of ~95% of all pups born (~17,000) in a matter of weeks, over 200 km of coastline along Península Valdés. Some newborns may have been infected before birth, as transplacental transmission of H5N1 HPAI viruses has been reported in humans⁵⁵ and high virus loads were detected in aborted sea lion fetuses^{11,30}. It could also be that mothers were infected and shed virus through their milk, infecting their pups^{56,57}. Yet, how adult female elephant seals would have been infected in the first place without mammal-to-mammal transmission presents a thornier question. Prior to arriving to give birth at Península Valdés in September, the elephant seals would have spent a solitary winter at sea in the South Atlantic and Southern Oceans⁵⁸. Feeding is an unlikely route of exposure to H5N1 HPAI viruses, since elephant seals do not eat birds or mammals, feeding instead on squid, fish and crustaceans captured in deep waters^{51,59}, and adult elephant seals will fast while on land^{51,60}. Moreover, elephant seals are pelagic and only come to shore and aggregate for a few weeks to breed and later to molt, thus limiting the time window for interspecific interactions and transmission on land^{51,61}. The main interactions between birds and elephant seals involve opportunistic scavenging of elephant seals' placental remains, molted skin and carcasses by gulls⁶⁰ (Supplementary

Figure 1E), which provides more opportunities for mammal-to-bird transmission than vice-versa. In this context, it seems unlikely that bird-to-mammal transmission alone could explain the simultaneity, extent and speed of the outbreak in elephant seals at Península Valdés. Although there are still many unknowns about the precise viral transmission routes (e.g., contact, environmental, aerosol), seal-to-seal transmission seems the most plausible hypothesis to explain viral dissemination during this outbreak. In experimental infections in several wild mammals and in ferret models, nasal and oral H5N1 virus shedding has occurred as quickly as one day post-inoculation and lasted about a week⁶², suggesting that elephant seals infected soon after hauling out on the beach could have begun shedding virus within a very short period of time.”

Additionally, we discuss what data would be necessary to corroborate/dismiss the hypothesis of mammal-mammal transmission in the following section:

“Despite gaps in the available data, our epidemiological and phylogenetic results support the hypothesis that the spread of viruses from the novel marine mammal clade in South America has occurred via mammal-to-mammal transmission. And while there is a need for better understanding the mode of transmission between marine mammals, there is increasing consensus that mammal-to-mammal transmission has played a significant role in the recent spread of H5N1 HPAI viruses worldwide⁷⁸. The recovery of live virus for pathogenesis and transmission studies would be valuable to demonstrate how these strains behave in mammalian experimental models. Furthermore, additional studies on the virus prevalence, shedding, and genome in different potential hosts within the coastal wildlife of Patagonia, especially for species that were already shown to be susceptible to H5N1 HPAI viruses in other regions (e.g., skuas, gulls, petrels) would be helpful to identify if there are asymptomatic reservoirs of infection and bridge hosts.”

Please also address the following minor comments.

1. Please add back in that which HA numbering you are using when referring to A133S.

RESPONSE:

The manuscript and the legend of Figure 3 and Supplementary Figure 9 have been modified to indicate that the A133S mutation in HA refers to the H5 numbering, as suggested by the reviewer.

2. It is still unclear to me how and when you pooled samples. Pooling according to sample type is not clear to me. Please make it clear whether you pooled all the samples from a single individual to test for influenza or across multiple individuals. Were any pooled samples then separated for sequencing?

RESPONSE:

Brain swabs of the same species, site and date were pooled for testing (for example, brain swabs from elephant seal pups collected at Punta Delgada at 10-Oct-2023 were pooled for initial screening). Alternatively, all samples from different swabs of the same individual were pooled (for example, the samples from brain, cloaca, lung and orotracheal swabs of individual CH-PD037 were pooled for

initial screening). Please note that Supplementary Table 2 provides a detailed breakdown of how samples were pooled for testing on an assay-by-assay basis, as well as the CT value for each test. The manuscript has also been clarified in results and methods (changes tracked in the manuscript).

3. L256 please remove “clearly”.

RESPONSE:

Done.

4. L286. I would find it very helpful if the authors could give some information on the timeline of infections that they are envisioning given what is known about incubation time of the virus. This would help demonstrate that most infections likely occurred after haul-out.

RESPONSE:

The incubation period for HPAI in elephant seals is unknown. However, a recent experimental study on a number of wild mammals and in ferret models shows that oral and nasal shedding may occur as quickly as one day post-inoculation (Root et al. 2024; <https://doi.org/10.1016/j.virol.2024.110231>). Assuming that a similar dynamic occurs in elephant seals, the incubation time after haul-out would not have been a limiting factor for the rapid spread of the infection among seals. We have edited the manuscript to point to this recent data, complementing our discussion about the hypothesized chronology of events (changes tracked in the manuscript).